# Drop-in Circulant Structural Priors for Transformer Decoding of Cyclic Codes

**Shuai Xiao** [1 2 3]   **Weijun Fang** [4 3]   **Qiaosheng Zhang** [5 6]

## Abstract

While Transformer-based architectures have revolutionized neural decoding, existing models often treat codes as generic sequences, ignoring their inherent algebraic properties. In this paper, we take a step toward bridging these two domains by proposing a decoding approach that integrates the algebraic structure of cyclic codes into Transformer-based decoders. Building on coding theory, we introduce two key notions, *error correction patterns* and *inter-node relationships*, and show how they can be exploited in neural architectures. By further leveraging the inherent cyclic properties of these codes, we propose a plug-and-play, flexibly deployable decoding method tailored for cyclic codes, which links the structural characteristics of the codes to the model parameters. Experimental results show that our method reduces the bit error rate (BER) by about one order of magnitude on average, while also reducing the total number of parameters by approximately 97%. Additional comparative experiments provide evidence supporting our proposed notions and highlight a promising pathway for bridging classical coding theory and modern Transformer-based decoding architectures.

## 1. Introduction

In modern digital communication, error correction codes (ECCs) are specifically designed to introduce redundancy through information encoding, enabling the detection and correction of transmission errors and thereby enhancing communication reliability. A fundamental challenge lies in developing decoders that approach, or ideally achieve, the theoretically optimal performance of maximum-likelihood decoding, which is known to be an NP-hard problem. Recent advancements in deep learning have led to the emergence of neural network-based decoders (Nachmani et al., 2016; Lugosch & Gross, 2017; Nachmani et al., 2018; Nachmani & Wolf, 2019; Buchberger et al., 2021), particularly model-free neural decoders employing generic neural architectures (Cammerer et al., 2017; Gruber et al., 2017; Kim et al., 2018). These data-driven approaches eliminate the reliance on predefined decoding algorithms, and have demonstrated superior flexibility and performance compared to conventional methods.

Among model-free neural decoders, Transformer-based architectures have gained significant attention. Originally developed for natural language processing, the self-attention mechanism in Transformers effectively captures long-range dependencies among input elements and has shown exceptional efficacy across multiple domains. This line of work led to the Error Correction Code Transformer (ECCT) (Choukroun & Wolf, 2022), which incorporates a mask matrix derived from the parity-check matrix (PCM) to explicitly model inter-codebit relationships, and demonstrates remarkable decoding capabilities. Building on this, FECCT (Choukroun & Wolf, 2024a) and MM-ECCT (Park et al., 2025) further enhance performance through refined characterization of codebit proximity relationships and by optimizing the PCM for a better construction of the mask matrix. The CrossMPT (Park et al., 2024) architecture further improves decoding efficiency by integrating message-passing mechanisms with cross-attention structures, enabling separate updates of magnitude and syndrome while achieving significant performance gains.

Despite the growing interest in Transformer-based decoders, much of the current research focuses on optimizing various performance indicators, with limited effort devoted to understanding the fundamental principles underlying their success. For instance, while embedding dimensions are conventionally understood as representing feature characteristics of positional elements, the precise definition of these features and the rationale behind performance gains with increased dimensions remain unclear. Similarly, the intrinsic significance of the parameter matrices in these models and

---

[1]Interdisciplinary Center, Shandong University, Qingdao, China [2]Research Center for Mathematics and Interdisciplinary Sciences, Shandong University, Qingdao, China [3]School of Cyber Science and Technology, Shandong University, Qingdao, China [4]State Key Laboratory of Cryptography and Digital Economy Security, Shandong University, Qingdao, China [5]Shanghai Innovation Institute, Shanghai, China [6]Shanghai Artificial Intelligence Laboratory, Shanghai, China. Correspondence to: Weijun Fang <fwj@sdu.edu.cn>.

*Proceedings of the 43rd International Conference on Machine Learning*, Seoul, South Korea. PMLR 306, 2026. Copyright 2026 by the author(s).

their inherent relationships with embedding dimensions lack rigorous explanations. Despite growing empirical success, theoretical interpretations remain relatively underdeveloped, and a more principled understanding is still an open direction.

Furthermore, most existing studies focus on universal decoders and often overlook code-specific properties, even though practical systems typically employ specific code families. Therefore, explicitly incorporating the inherent algebraic structures of codes into Transformer-based decoders is crucial. For example, neural belief propagation (BP) decoding has successfully leveraged cyclic invariance in cyclic codes (Chen & Ye, 2021) and quasi-cyclic structures in QC-LDPC codes (Zhang et al., 2024). Recently, preliminary applications of cyclic codes in Transformer-based decoders have also been proposed (Xiao et al., 2025).

To address the aforementioned limitations, we leverage coding theory and Transformer-based decoders to formally propose and investigate two core claims: (i) circulant (square) PCM reduces the diversity of Error Correction Patterns, and (ii) embedding dimensions intrinsically encode inter-node relationships. Building on the above claims, our reasoning and method are intuitively more natural and theoretically more logical than prior work (Xiao et al., 2025), and achieve superior performance. Our key contributions are as follows:

- By incorporating algebraic structure of cyclic codes, we propose a plug-and-play optimization approach for cyclic codes that reduces the bit error rate (BER) by about one order of magnitude and decreases parameter counts by over $97\%$ on average when applied to some mainstream Transformer-based decoders, including ECCT, CrossMPT, and MM-ECCT.

- We show that learned embeddings reflect Tanner-graph inter-node relationships, providing an interpretable, testable basis for code-aware parameter sharing.

- We introduce Error Correction Patterns and use them to guide the construction of a circulant PCM, suggesting a promising direction for improving neural decoders and PCM design.

Collectively, these contributions have the potential to bridge coding theory with neural decoding mechanics, helping to establish new principles for developing efficient, interpretable, and code-specialized decoding architectures.

## 2. Background

### 2.1. Error Correcting Codes and Cyclic Codes

An $[n, k]$ binary linear code $C$ is a $k$-dimensional subspace of $\mathbb{F}_2^n$. The code $C$ is called cyclic if, for every codeword $(a_0, a_1, \ldots, a_{n-1}) \in C$, its cyclic shift

$(a_{n-1}, a_0, a_1, \ldots, a_{n-2})$ is also in $C$. The generator matrix $G$ of $C$ is a $k \times n$ binary matrix whose rows span the code $C$. The PCM $H$ of $C$ is an $(n-k) \times n$ binary matrix such that $C = \{c \in \mathbb{F}_2^n \mid H \cdot c^\top = 0\}$. A message $m$ is encoded into a codeword $x \in C$ by multiplication with the generator matrix $G$ (i.e., $x = m \cdot G$, $m \in \{0,1\}^k$). Assuming transmission over an Additive White Gaussian noise (AWGN) channel, $x \in \{0,1\}^n$ is modulated into the transmitted signal $x_s$ using binary phase-shift keying (BPSK) modulation (i.e., over $\{\pm 1\}$). The channel output is $y = x_s + z$, where $z \sim \mathcal{N}(0, \sigma^2)$. The decoder operates by first calculating the syndrome $s(y) = Hy_b$, where $y_b = \text{bin}(\text{sign}(y))$ denotes the binarized hard-decision version of $y$. The function $\text{sign}(a)$ equals $+1$ if $a \geq 0$ and $-1$ otherwise, and $\text{bin}(a)$ maps $-1$ to 1 and $+1$ to 0. The decoder checks whether $s(y) = 0$ to determine if $y$ contains transmission errors induced by noise. If $s(y) \neq 0$, the decoder initiates error correction to recover the original transmitted codeword $x$.

### 2.2. Transformer-based Decoders

Transformer-based decoders (e.g., ECCT (Choukroun & Wolf, 2022)) typically employ syndrome-based pre-processing to mitigate the overfitting issue in model-free decoders (Bennatan et al., 2018), an issue that performs well on training codewords but generalizes poorly to unseen codewords. The specific procedure concatenates the received codeword $y$ with its syndrome $s(y)$ to form the augmented vector $\tilde{y} = [|y|, s(y)]$, where $|y|$ denotes the absolute value of $y$, which serves as the decoder input vector. Before decoding, each element of $\tilde{y}$ is embedded into a high-dimensional space to extract additional features, resulting in $\Phi = (\tilde{y} \otimes 1_d^\top) \odot \hat{W} \in \mathbb{R}^{(2n-k) \times d}$, where $\hat{W} \in \mathbb{R}^{(2n-k) \times d}$ is a learnable embedding matrix, $\otimes$ denotes the Kronecker product and $\odot$ denotes the Hadamard product.

For convenience, we rewrite $y = x_s + z$ in a multiplicative form as $y = x_s \tilde{z}_s$, where $\tilde{z}_s$ denotes the multiplicative noise. The decoder function $f$ maps $y$ to an estimate of the multiplicative noise (i.e., $f(y) = \hat{z}_s$), and then reconstructs the codeword as $\hat{x} = \text{bin}(\text{sign}(y f(y))) = \text{bin}(\text{sign}(x_s \tilde{z}_s \hat{z}_s))$. Note that $\hat{x} = x$ whenever $\text{sign}(\hat{z}_s) = \text{sign}(\tilde{z}_s)$.

The attention mechanism serves as the core component of Transformer-based decoders, comprising four key elements: Query ($Q$), Key ($K$), Value ($V$), and the mask matrix $M$. The $Q$, $K$, and $V$ matrices are obtained by projecting $\phi$ through distinct weight matrices: $Q = \Phi W^Q$, $K = \Phi W^K$, $V = \Phi W^V$, where $W^Q, W^K, W^V \in \mathbb{R}^{d \times d}$. The mask matrix $M$ is constructed based on the PCM of the code $C$, and its sparsity critically influences decoder performance — hence most current improvements focus on this aspect (Choukroun & Wolf, 2022; 2023; 2024a;b; Park et al., 2024; 2025). These matrices ultimately generate outputs through

the equation:

$$\text{Attention}_{\text{H}}(Q, K, V) = \text{Softmax}\left(\frac{QK^{\top} + M}{\sqrt{d}}\right)V. \quad (1)$$

In CrossMPT (Park et al., 2024), the cross-attention mechanism necessitates splitting the $\Phi$ into two distinct components that undergo separate updating processes.

## 3. Cyclic Codes Perspective on Transformer

### 3.1. Cyclic Equivalence in PCM

For a PCM $H$ of size $r \times n$, the rows and columns of $H$ are referred to as check nodes CNs $= \{c_1, c_2, \ldots, c_r\}$ and variable nodes VNs $= \{v_1, v_2, \ldots, v_n\}$, respectively (as shown in Figure 1a). Each check node corresponds to a parity-check equation involving some variable nodes. In cyclic codes, the check nodes already exhibit a degree of cyclic structure when using an $(n - k) \times n$ PCM $H$, as illustrated in Figure 1a. If we extend $H$ to a circulant PCM, this cyclicity is expressed more fully—both the check nodes and the variable nodes then possess cyclic structure, such as Figure 1b.

#### 3.1.1. WHY A CIRCULANT PCM

**Definition 3.1** (Error Correction Patterns). Given a PCM $H$, for each variable node $v_i$, we define its error correction pattern $\text{ECP}(v_i)$ as the set of the parity-check equations that includes $v_i$.

For instance, the PCM of the $(7, 4)$ Hamming code is shown in Figure 1a. The error correction patterns of its variable nodes are as follows: $\text{ECP}(v_1) = \{c_1\}$, $\text{ECP}(v_2) = \{c_2\}$, $\text{ECP}(v_3) = \{c_1, c_3\}$, $\text{ECP}(v_4) = \{c_1, c_2\}$, $\text{ECP}(v_5) = \{c_1, c_2, c_3\}$, $\text{ECP}(v_6) = \{c_2, c_3\}$, $\text{ECP}(v_7) = \{c_3\}$. The $(7, 4)$ Hamming code is a cyclic code, and the check nodes $c_2$ and $c_3$ are the cyclic shifts of $c_1$ by one and two positions, respectively. Therefore, we divide the error correction patterns of these variable nodes into 4 classes according to the cyclic shift properties of their associated check nodes: $\{\text{ECP}(v_1), \text{ECP}(v_2), \text{ECP}(v_7)\}$, $\{\text{ECP}(v_4), \text{ECP}(v_6)\}$, $\{\text{ECP}(v_3)\}$, and $\{\text{ECP}(v_5)\}$. $\text{ECP}(v_3)$ and $\text{ECP}(v_4)$ do not belong to the same class, since the cyclic shift of $c_3$ is not equal to $c_1$.

In the previous ECCT, CrossMPT, and MM-ECCT models, the PCMs were usually chosen to have the conventional size of $(n - k) \times n$. However, as seen in the example above, this approach leads to a somewhat complicated partition of the error correction patterns. To mitigate this, we extend the PCM to size $n \times n$ (i.e., circulant/square matrix) by applying cyclic shifts. As shown in Figure 1b, the PCM of the $(7, 4)$ Hamming code becomes a $7 \times 7$. The error correction patterns of its variable nodes are then: $\text{ECP}(v_1) = \{c_1, c_4, c_5, c_6\}$, $\text{ECP}(v_2) = \{c_2, c_5, c_6, c_7\}$, $\text{ECP}(v_3) = \{c_1, c_3, c_6, c_7\}$, $\text{ECP}(v_4) = \{c_1, c_2, c_4, c_7\}$,

$\text{ECP}(v_5) = \{c_1, c_2, c_3, c_5\}$, $\text{ECP}(v_6) = \{c_2, c_3, c_4, c_6\}$, $\text{ECP}(v_7) = \{c_3, c_4, c_5, c_7\}$. It can be observed that all these error correction patterns are equivalent under cyclic shifts. In general, we have:

*Conclusion* 3.2. Let $C$ be a binary cyclic code, and let $H$ denote its PCM, which is an $n \times n$ circulant matrix (whose rows are cyclic shifts of the first row). Then, the error correction patterns associated with the variable nodes are equivalent under cyclic shifts.

*Proof.* For any $1 \leq j \leq n$, by the definition of Error Correction Patterns, $c_i \in \text{ECP}(v_j) \iff H_{i,j} = 1$. Since the next row of $H$ is a right cyclic shift of the previous row, we have $H_{i,j} = 1 \iff H_{i+1,j+1} = 1$, where the indices are taken modulo $n$. By the definition again, $c_i \in \text{ECP}(v_j) \iff c_{i+1} \in \text{ECP}(v_{j+1})$. Thus, $\text{ECP}(v_{j+1}) = \{c_{i+1} : c_i \in \text{ECP}(v_j)\}$, namely, $\text{ECP}(v_{j+1})$ is the right cyclic shift of $\text{ECP}(v_j)$.

Therefore, after adopting the $n \times n$ circulant PCM, we observe that the complete cyclic property obtained through row-wise cyclic shifts also extends to the column dimension. At this point, column-wise cyclic symmetry implies that all variable nodes share a unified error correction pattern, equivalent to $\text{ECP}(v_1)$. We expect that the model will then focus more on learning this single common correction pattern. Moreover, as shown in Figure 1c, small square matrices derived from the PCM exhibit the same cyclic properties. Even when these matrices are treated as individual nodes, the positional invariance and cyclic equivalence of the correction patterns still hold.

#### 3.1.2. INTER-NODE RELATIONSHIP

In Transformer-based decoders, four types of inter-node relationships are typically considered: variable-to-variable (V-V), check-to-variable (C-V), variable-to-check (V-C), and check-to-check (C-C).

For C-V relationships, the connection from a check node $c_i$ to VNs is denoted as $c_i$-VNs$= \{v_{i_1}, v_{i_2}, \ldots, v_{i_r}\}$, indicating that $v_{i_1}, v_{i_2}, \ldots, v_{i_r}$ are directly connected to $c_i$. For example, taking the $(7, 4)$ Hamming code with a circulant PCM as in Figure 1b, $c_1$-VNs$=\{v_1, v_3, v_4, v_5\}$, $c_2$-VNs$=\{v_2, v_4, v_5, v_6\}$. Then $c_2$-VNs is a cyclic shift of $c_1$-VNs. The same applies analogously to V-C relationships.

For V–V relationships, there is no direct connection between a variable node and other variable nodes; the connection must be established through check nodes. Then, the connection from a variable node $v_i$ to VNs is denoted as $v_i$–VNs $= \{c_{i_1}$–VNs$, c_{i_2}$–VNs$, \ldots, c_{i_r}$–VNs$\}$, where $\{c_{i_1}, c_{i_2}, \ldots, c_{i_r}\} = v_i$–CNs, indicating that $v_i$ through its directly connected check nodes $v_i$–CNs, is indirectly connected to other variable nodes $\{c_{i_1}$–VNs$, c_{i_2}$–VNs$, \ldots, c_{i_r}$–VNs$\}$.

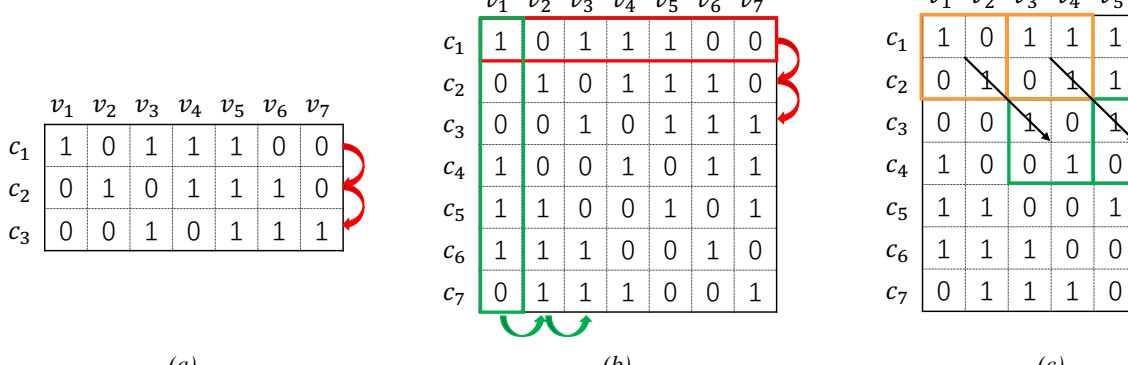

*Figure 1.* (a) The PCM of the (7,4) Hamming code; (b) its square extension via a cyclic shift of the first row; (c) an illustration of a cyclic shift of a small square matrix.

For instance, $v_1$–VNs=$\{c_1$-VNs, $c_4$-VNs, $c_5$-VNs, $c_6$-VNs$\}$

$= \{\{v_1, v_3, v_4, v_5\}, \{v_1, v_4, v_6, v_7\}, \{v_1, v_2, v_5, v_7\}, \{v_1, v_2, v_3, v_6\}\}$, $v_2$–VNs=$\{\{v_2, v_4, v_5, v_6\}, \{v_1, v_2, v_5, v_7\},$

$\{v_1, v_2, v_3, v_6\}, \{v_2, v_3, v_4, v_7\}\}$. Then $v_2$–VNs is a cyclic shift of $v_1$–VNs. The same applies analogously to C-C relationships. In general, we have:

*Conclusion* 3.3. For a binary cyclic code with a circulant PCM, the relationships $c_i$–VNs, $v_i$–CNs, $v_i$–VNs and $c_i$–CNs are cyclic shifts of $c_1$–VNs, $v_1$–CNs, $v_1$–VNs and $c_1$–CNs, respectively.

*Proof.* By the definition, $v_j \in c_1$–VNs $\iff c_1 \in \text{ECP}(v_j)$. By Conclusion 3.2, we have $c_1 \in \text{ECP}(v_j) \iff c_i \in \text{ECP}(v_{j+i-1}) \iff v_{j+i-1} \in c_i$–VNs, where the indices are taken modulo $n$. Thus $c_i$–VNs=$\{v_{j+i-1} : v_j \in c_1$–VNs$\}$.

Note that $v_i$–CNs=ECP($v_i$), the conclusion then follows from Conclusion 3.2.

$c_j$–VNs $\in v_1$–VNs $\iff v_1 \in c_j$–VNs. By the above, $v_1 \in c_j$–VNs $\iff v_i \in c_{j+i-1}$–VNs $\iff c_{j+i-1}$–VNs $\in v_i$–VNs. Thus $v_i$–VNs=$\{c_{j+i-1}$–VNs : $c_j$–VNs $\in v_1$–VNs$\}$

$v_j$–CNs $\in c_1$–CNs $\iff c_1 \in v_j$–CNs. By the above, $c_1 \in v_j$–CNs $\iff c_i \in v_{j+i-1}$–CNs $\iff v_{j+i-1}$–CNs $\in c_i$–CNs. Thus $c_i$–CNs=$\{v_{j+i-1}$–CNs : $v_j$–CNs $\in c_1$–CNs$\}$

The conclusion above demonstrates that inter-node relationships V-V, C-V, V-C and C-C can all be fully represented by cyclic shifts of $v_1$-VNs, $c_1$-VNs, $v_1$-CNs, and $c_1$-CNs, respectively. This significantly reduces the relational complexity across all nodes. Building on this observation, we propose several optimizations in the following chapter.

### 3.2. Explainable Embedding mechanism

When an $n \times n$ circulant PCM is used, the length of the parity-check part in the input vector becomes $n$, and the overall input length is $2n$. Consequently, the embedding matrix becomes $\Phi \in \mathbb{R}^{2n \times d}$. It has been empirically observed that increasing the embedding dimension $d$ improves the decoding performance, but the underlying reason remains unclear. Our motivation comes from the attention matrix $QK^\top$, which we decompose into individual components: let $\Phi = [\phi_1^\top, \phi_2^\top, \ldots, \phi_{2n}^\top]^\top$, where $\phi_i \in \mathbb{R}^{1 \times d}$ denotes the $i$-th row of $\Phi$; let $W^Q = [W_1^Q, W_2^Q, \ldots, W_d^Q]$ and $W^K = [W_1^K, W_2^K, \ldots, W_d^K]$, where $W_j^Q$ and $W_j^K \in \mathbb{R}^{d \times 1}$ denote the $j$-th column vectors of $W^Q$ and $W^K$, respectively. Then we have $Q = [(\phi_1 W^Q)^\top, (\phi_2 W^Q)^\top, \ldots, (\phi_{2n} W^Q)^\top]^\top$, and $K = [(\phi_1 W^K)^\top, (\phi_2 W^K)^\top, \ldots, (\phi_{2n} W^K)^\top]^\top$. We can then write the attention matrix as follows: $QK^\top =$

$$
\begin{bmatrix}
\phi_1 W^Q (W^K)^\top \phi_1^\top & \phi_1 W^Q (W^K)^\top \phi_2^\top & \cdots & \phi_1 W^Q (W^K)^\top \phi_{2n}^\top \\
\phi_2 W^Q (W^K)^\top \phi_1^\top & \phi_2 W^Q (W^K)^\top \phi_2^\top & \cdots & \phi_2 W^Q (W^K)^\top \phi_{2n}^\top \\
\vdots & \vdots & \ddots & \vdots \\
\phi_{2n} W^Q (W^K)^\top \phi_1^\top & \phi_{2n} W^Q (W^K)^\top \phi_2^\top & \cdots & \phi_{2n} W^Q (W^K)^\top \phi_{2n}^\top
\end{bmatrix}
\tag{2}
$$

From the equation above, we observe that the $(i, j)$-th entry of the attention matrix is determined by $\phi_i$ and $\phi_j$, reflecting the relational proximity between node $i$ and node $j$ (either variable node or check node). Since the magnitude or syndrome components in $\phi_i$ or $\phi_j$ remain fixed, this relationship is governed by their respective embedding vectors. Inspired by this observation, we propose a hypothesis regarding the role of embedding dimensions in neural decoders. We suggest that, unlike in NLP where dimensions represent abstract semantic features, in the context of decoding, embedding dimensions may serve as carriers for topological information of the code's Tanner graph; specifically, the embedding vector $\phi_i$ encodes the relational patterns between node $i$ and all other nodes.

In contrast to the standard practice of treating $d$ as a hyperparameter tuned for capacity, we propose a novel and

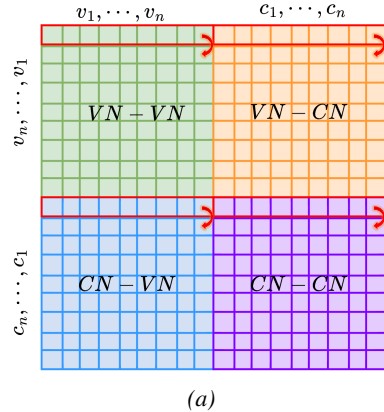

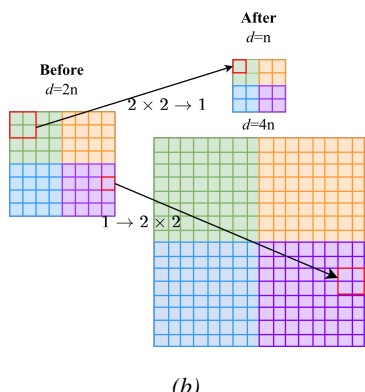

*(a)* *(b)*

*Figure 2.* (a) Block-wise cyclic shifting applied to the transposed parameter matrix; and (b) the corresponding structural adaptations when the embedding dimension is modified — $r = \frac{1}{2}$ and $r = 2$.

principled interpretation defined as:

$$
\hat{\phi}_i = \begin{cases} |y_i|\hat{W}_i, & \text{if } i \leq n, \\ \left(1 - 2(s(y))_{i-n}\right)\hat{W}_i, & \text{otherwise}, \end{cases} \tag{3}
$$

where $\{\hat{W}_i \in \mathbb{R}^{2rn}\}_{i=1}^{2n}$, and $r$ is a positive rational number such that the resulting $d = 2rn$ is a positive integer. Since the input to the decoder has length $2n$, it corresponds to $2n$ nodes in the Tanner graph: the first $n$ positions correspond to variable nodes, and the last $n$ positions correspond to check nodes. Specifically, we explore the distinct case where the embedding dimension $d$ matches the total number of nodes in the Tanner graph (i.e., $d = 2n$). In this setting, we conjecture that the network can learn a one-to-one mapping (bijection) where the $k$-th dimension of the embedding vector is consistent with the connectivity to the $k$-th node in the graph. This transforms the embedding space from a latent feature space into a structured adjacency representation. Moreover, due to the cyclic equivalence of these relations discussed in Section 3.1.2, we can scale the embedding dimension as $d = 2rn$ and reconstruct $\hat{W}$ through cyclic shifts of certain representative nodes. The details are given in Section 3.3 and Appendix A.

While this interpretation is heuristic, our empirical results in Appendices D and E (see Figures 11 and 12) provide strong evidence supporting this hypothesis, showing a clear saturation of performance as $d$ approaches $2n$.

### 3.3. Cyclic Parameter Reuse Mechanism

For clarity, we first consider the case $d = 2n$. In this case, each attention parameter matrix (e.g., $W^Q$) and the embedding output $\Phi$ has size $2n \times 2n$. Following the same

rationale as in Equation (2), we expand $Q = \Phi W^Q$ as

$$
Q = \begin{bmatrix} \phi_1 W_1^Q & \phi_1 W_2^Q & \dots & \phi_1 W_d^Q \\ \phi_2 W_1^Q & \phi_2 W_2^Q & \dots & \phi_2 W_d^Q \\ \vdots & \vdots & \ddots & \vdots \\ \phi_{2n} W_1^Q & \phi_{2n} W_2^Q & \dots & \phi_{2n} W_d^Q \end{bmatrix}. \tag{4}
$$

From a column-wise perspective, Equation (4) shows that each column $W_i^Q$ defines how the $i$-th embedding dimension is linearly combined across all positions. Combined with our interpretation of $\phi_i$ in Section 3.2, this means that the parameter matrix controls how the inter-node relationships encoded in the embeddings are further processed. Then we deduce that $[W_1^Q, W_2^Q, \ldots, W_{2n}^Q]$ should also exhibit the inter-node relationship. In light of the symmetry induced by the mask matrix, we can also view the parameter matrix row-wise. That is, the $i$-th row or column vector encodes the relationships between the $i$-th node and all $2n$ nodes.

Because the PCM is circulant and the inter-node relationships are cyclically equivalent (Conclusion 3.3), these rows should not be learned independently. Instead, they are expected to exhibit cyclic-shift structure with respect to the node index. In particular, for $d = 2n$, there is a bijection between the nodes and the dimensions. So it can be sufficient in our setting to learn two representative rows: a representative variable-node row, say the row corresponding to node $v_1$ (with index 1), and a representative check-node row, say the row corresponding to node $c_1$ (with index $n+1$). All remaining rows can then be obtained by cyclically shifting these two representatives along the node index, separately for the first $n$ columns (variable nodes) and the last $n$ columns (check nodes), as illustrated in Figure 2a.

For the attention parameter matrices (e.g., $W^Q$), starting from dimension $d = 2n$, we can scale it to $d = 2rn$, where $r$ is a positive rational number such that the resulting $d = 2rn$

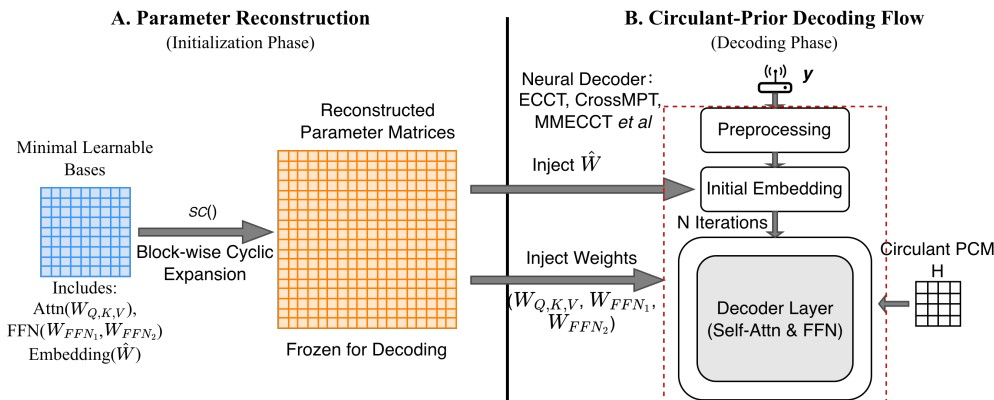

*Figure 3.* Illustration of the plug-and-play method. The most novel aspect lies in preserving the original decoding logic and architecture of Transformer-based decoders while introducing cyclic reuse mechanisms for both embedding and parameter matrices (the function sc()), while $H$ is an $n \times n$ circulant matrix. It is worth mentioning that our method can be applied to all parameter matrices, such as $\hat{W}, W^Q, W^K, W^V$, and can also be extended to the feed-forward network by adopting cyclic-shift rules similar to those above (the details are given in Appendix B).

is a positive integer, thereby obtaining a $2rn \times 2rn$ matrix. In this case, the relationships can be described as follows (as shown in Figure 2b):

- **Dimension Reduction ($d \downarrow, r < 1$):** Each group of $\frac{1}{r} \times \frac{1}{r}$ scalar relationships is collapsed into a single representation, while we keep the same number of representative rows (e.g., $d = 2n \xrightarrow{r=\frac{1}{2}} n$, maintain 2 representative rows (one variable node, one check node)).

- **Dimension Expansion ($d \uparrow, r > 1$):** A single relationship expands into $r \times r$ representations, which necessitates additional representative rows (e.g., $d = 2n \xrightarrow{r=2} 4n$, with 4 representative rows (two variable node, two check node)).

Therefore, based on the definitions of cyclic equivalence in node relationships (Section 3.1.2), the cyclic dependencies inherent in embedding and parameter matrices (Section 3.2 and this section) and the dimension scaling method, we propose the following parameter matrix reconstruction method:

$$W = \begin{cases} \text{sc}(1, \frac{n+1}{r}), & \text{if } r \leq 1, \\ \text{sc}(1, \dots, r, rn+1, \dots, rn+r), & \text{if } r > 1, \end{cases} \quad (5)$$

Here, $(1, \dots, r)$ and $(rn+1, \dots, rn+r)$ denote the indices of the representative rows for the variable nodes and the check nodes, respectively (the same applies to $r$ and $\frac{n+1}{r}$); the function $\text{sc}(\cdot)$, described in Appendix A, reconstructs the parameter matrix by performing block-wise cyclic shifting on these representative rows; and $W$ represents the transpose of $\hat{W}, W^Q, W^K, W^V$. As illustrated in Figure 2a, for

$d = 2n$, the cyclic shifting preserves structural consistency. In addition, for $\hat{W}$, the scaling rules and reconstruction procedure differ slightly from those for the standard attention parameter matrices, and we will explain these differences in Appendix A.

In summary, from the perspective of coding theory, we provide a clear interpretation of the embedding vectors and, building on this viewpoint, propose a cyclic reconstruction method for the parameter matrices. Next, we present experimental results to validate both the effectiveness and the interpretability of the proposed method.

### 3.4. Architecture and Training

As shown in Figure 3, the proposed method can be readily adapted to a variety of Transformer-based decoders. We adopt the same training setup used in the previous work (Choukroun & Wolf, 2022; 2023; 2024a;b; Park et al., 2024; 2025): we train for 1000 epochs, with 1000 minibatches per epoch and 128 samples per minibatch, using the Adam optimizer for a fair comparison. The learning rate is initially set to $10^{-4}$ and is gradually decreased to $5 \times 10^{-7}$ following a cosine decay schedule. As in (Bennatan et al., 2018), training on the all-zero codeword is sufficient for SNRs($E_b/N_0$) from 3 dB to 7 dB. All experiments were conducted on an NVIDIA GeForce RTX 4090 GPU and an AMD EPYC 7402 CPU.

## 4. Experiments

To validate the theoretical soundness of the proposed method and its empirical validity in practical experiments, we conducted experiments on two classical cyclic code families: Bose-Chaudhuri-Hocquenghem (BCH) codes (Bose & Ray-

*Table 1.* We implement our method in ECCT, CrossMPT, and MM-ECCT to compare the negative natural logarithm of BER (i.e. -ln(BER)) improvements, and higher is better. It is critical to note that for the systematic portion of MM-ECCT, we only zero-pad the systematic PCM into square size , and also apply the parameter reuse mechanism in this part. In tabular results, we employ three embedding dimensions $(d = n, 2n, 4n)$ to validate our theoretical definitions of embedding dimensionality and their scaling mechanisms. At $SNR = 6$, our method reduces the BER by an order of magnitude on average.

| Decoder | CE BP | | | ECCT | | | CrossMPT | | | MM-ECCT | | | ECCT Ours | | | CrossMPT Ours | | | MM-ECCT Ours | | |
|---|---|---|---|---|---|---|---|---|---|---|---|---|---|---|---|---|---|---|---|---|---|
| Code/SNR | 4 | 5 | 6 | 4 | 5 | 6 | 4 | 5 | 6 | 4 | 5 | 6 | 4 | 5 | 6 | 4 | 5 | 6 | 4 | 5 | 6 |
| BCH(63,36) | 4.65 | 6.35 | 8.72 | 4.41 | 6.1 | 8.56 | 4.63 | 6.46 | 9.04 | 5.58 | 7.75 | 10.87 | **4.87** | **7.02** | **10.16** | 5.34 | 7.76 | 11.33 | 5.65 | 7.79 | 11.31 |
| | | | | 4.69 | 6.57 | 9.18 | 4.86 | 6.82 | 9.64 | | | | **5.13** | **7.40** | **10.71** | 5.47 | 7.93 | 11.75 | | | |
| | | | | 5.02 | 7.06 | 10.02 | 5.02 | 7.09 | 9.99 | | | | **5.44** | **7.93** | **11.43** | 5.60 | 8.11 | 11.80 | | | |
| BCH(63,45) | 5.12 | 6.96 | 9.46 | 5.01 | 7.01 | 9.77 | 5.23 | 7.35 | 10.28 | 5.93 | 8.41 | 11.67 | **5.72** | **8.33** | **11.98** | 6.02 | 8.94 | 12.71 | 6.12 | 8.88 | 12.85 |
| | | | | 5.12 | 7.19 | 9.92 | 5.41 | 7.61 | 10.8 | | | | **5.85** | **8.63** | **12.41** | 6.19 | 9.21 | 13.19 | | | |
| | | | | 5.79 | 8.15 | 11.60 | 5.54 | 7.79 | 11.26 | | | | **6.14** | **9.03** | **12.94** | 6.36 | 9.42 | 13.49 | | | |
| BCH(63,51) | | | | 5.42 | 7.48 | 10.44 | 5.57 | 7.82 | 10.92 | 5.95 | 8.41 | 11.73 | **6.12** | **8.85** | **12.63** | 6.37 | 9.34 | 13.15 | 6.31 | 9.05 | 12.86 |
| | | | | 5.36 | 7.39 | 10.08 | 5.71 | 8.08 | 11.51 | | | | **6.26** | **9.06** | **12.86** | 6.52 | 9.35 | 13.17 | | | |
| | | | | 5.31 | 7.22 | 9.82 | 5.91 | 8.50 | 11.87 | | | | **6.48** | **9.40** | **13.15** | 6.52 | 9.35 | 13.27 | | | |
| PRM(63,42) | 5.92 | 8.26 | 10.85 | 4.76 | 6.47 | 8.86 | 4.94 | 6.91 | 9.67 | 6.39 | 9.20 | 12.91 | **5.44** | **7.78** | **11.07** | 5.48 | 7.85 | 11.36 | 6.47 | 9.33 | 13.13 |
| | | | | | | | 5.13 | 7.26 | 10.27 | | | | | | | 5.71 | 8.34 | 12.11 | | | |
| | | | | | | | 5.38 | 7.57 | 10.57 | | | | | | | 5.82 | 8.58 | 12.23 | | | |
| PRM(127,64) | 3.14 | 4.17 | 5.93 | 3.75 | 5.53 | 8.36 | 3.48 | 4.92 | 7.33 | | | | **3.98** | **6.00** | **9.18** | 3.64 | 5.37 | 8.29 | | | |
| | | | | | | | 3.75 | 5.53 | 8.47 | | | | | | | 4.02 | 6.17 | 9.83 | | | |
| PRM(127,99) | 5.69 | 8.19 | 11.15 | 5.06 | 7.54 | 11.33 | 5.15 | 7.72 | 11.55 | | | | **6.28** | **10.08** | **15.66** | 6.65 | 10.63 | 15.97 | | | |
| | | | | | | | 5.33 | 8.01 | 12.22 | | | | | | | 6.87 | 10.96 | 16.61 | | | |

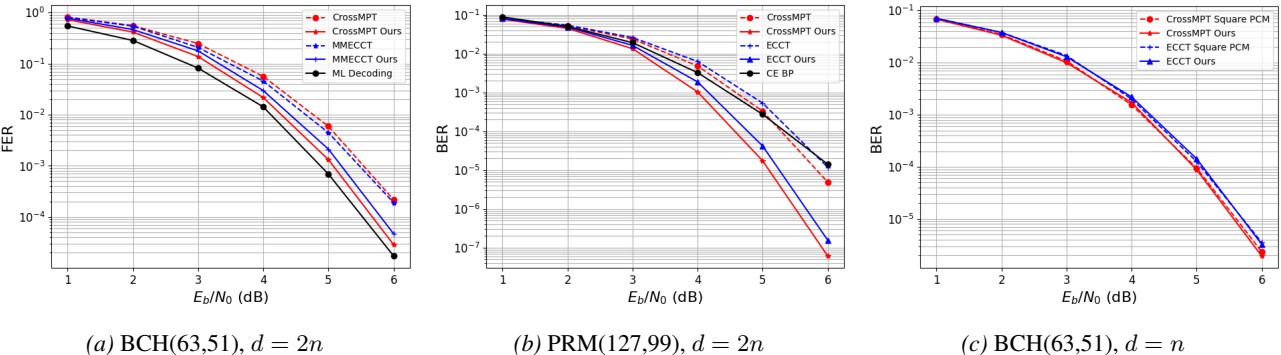

*(a)* BCH(63,51), $d = 2n$      *(b)* PRM(127,99), $d = 2n$      *(c)* BCH(63,51), $d = n$

*Figure 4.* (a) The Frame Error Rate (FER) performance of our method used in ECCT, CrossMPT and compared with ML decoding; (b) the BER performance of our method used in ECCT, CrossMPT; (c) comparing the variation of using parameter reuse with both use circulant (square) PCM in decoders.

Chaudhuri, 1960) and Punctured Reed-Muller (PRM) codes. Experimental results were evaluated using standard benchmark metrics, including bit error rate (BER) and its negative natural logarithm. All Transformer-based decoder experimental data presented in this work were obtained under identical configurations (e.g. number of decoder layer(s) $N = 6$) and PCM. Since Transformer-based decoders have already been sufficiently compared with numerous neural decoders to validate their superior performance, we solely added CE BP (Chen & Ye, 2021) (Cyclically Equivariant Neural Decoders for Cyclic Codes), a neural belief propagation (NBP) decoder optimized for cyclic codes, as the baseline.

### 4.1. Performance and Ablation Study

The experimental results presented in Table 1 illustrate that our method can be easily deployed on mainstream Transformer-based decoders (ECCT, CrossMPT, and MM-ECCT), leading to significant performance improvements. In particular, the gains are particularly pronounced for high-rate codes. For example, as shown in Figures 4a and 4b, our method used in the CrossMPT decoder achieves a nearly 0.8 dB improvement for PRM(127,99) codes, and even approaches Maximum Likelihood (ML) decoding in BCH(63,51), where the ML performance curves are taken from (Helmling et al., 2025). Further comparisons on performance are given in the Appendix C. Although such substantial gains might initially suggest alignment with the concept in Section 3.1.1, that reducing the diversity of error correction patterns drives the improvement—we designed additional validation experiments below.

In Table 2, we expanded the PCM into a square matrix by randomly selecting rows from the original $(n - k) \times n$ PCM

*Table 2.* Within the CrossMPT, we evaluate the validity of cyclic square matrix implementations. Here, R-select denotes a baseline method that expands the original $(n-k) \times n$ PCM into an $n \times n$ square matrix by randomly selecting rows and applying linear combinations.

| | SNR | BCH(63,45) | BCH(63,51) | PRM(63,42) | PRM(127,64) | PRM(127,99) | POLAR(64,32) | POLAR(128,86) | LDPC(121,80) |
|---|---|---|---|---|---|---|---|---|---|
| CrossMPT | 4 | 5.41 | 5.71 | 5.13 | 3.75 | 5.33 | 6.85 | 8.14 | 7.97 |
| | 5 | 7.61 | 8.08 | 7.26 | 5.53 | 8.01 | 9.54 | **12.01** | 12.56 |
| | 6 | 10.80 | 11.51 | 10.27 | 8.47 | 12.22 | 12.85 | **16.49** | 18.34 |
| CrossMPT-Rselect | 4 | 5.86 | 6.23 | 5.65 | 3.75 | 5.48 | **6.90** | **8.16** | **8.01** |
| | 5 | 8.36 | 9.07 | 8.21 | 5.51 | 8.32 | **9.58** | 11.66 | **12.65** |
| | 6 | 12.05 | 12.94 | 11.64 | 8.37 | 12.65 | **12.91** | 16.09 | **18.51** |
| CrossMPT Ours | 4 | **6.19** | **6.52** | **5.71** | **4.02** | **6.87** | | | |
| | 5 | **9.07** | **9.35** | **8.34** | **6.17** | **10.96** | | | |
| | 6 | **13.19** | **13.17** | **12.11** | **9.83** | **16.61** | | | |

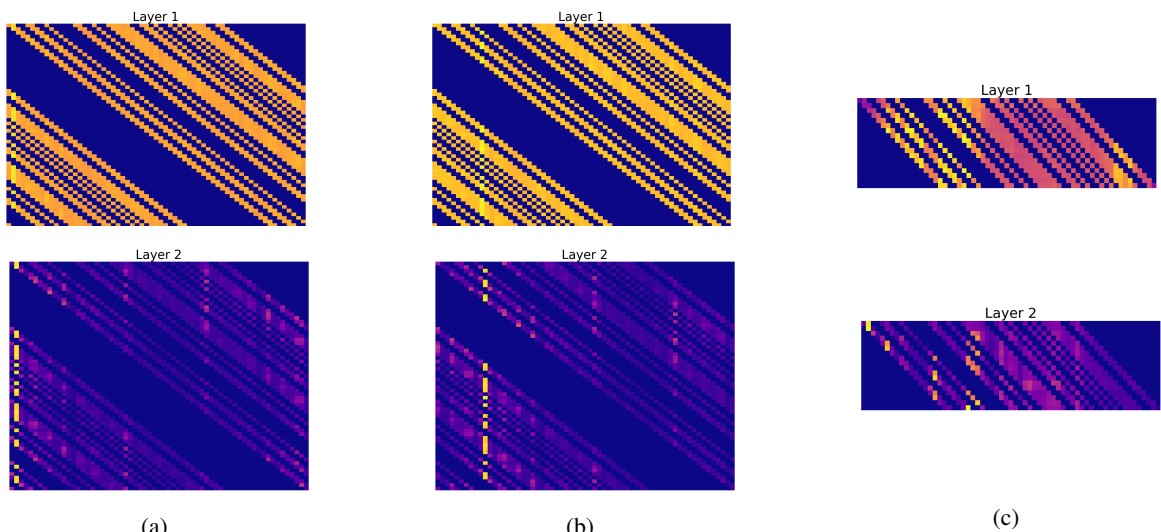

*Figure 5.* (a) and (b) show the masked attention matrix when decoding BCH(63, 45) using our method, with a single bit error at the second and 11th positions, respectively. (c) shows the result using standard CrossMPT with a bit error at the second position. Brighter areas indicate higher attention.

and applying linear combinations, aiming to test whether similar performance gains would persist. Our rationale stems from the observation that cyclic expansions inherently generate linear combinations of the first $n-k$ rows. The results in Table 2 reveal that for cyclic codes, even randomly expanded square matrices yield performance improvements, though these are inferior to those achieved by cyclic shift-based expansions. In contrast, for non-cyclic codes (e.g., POLAR codes, LDPC codes), random expansions may have little improvement or even degrade performance. These findings provide supporting evidence for our conjecture: circulant (square) matrices unify all error correction patterns under cyclic equivalence, rendering error-occurrence locations less critical to decoding success, while random expansions for cyclic codes retain partial cyclic properties (e.g., local row-wise cyclicity) to provide residual gains. For non-cyclic codes, expanding PCMs with random linear combinations introduces heterogeneous constraints, which may increase the complexity of error correction patterns the model must learn, and could hurt performance. Thus, cyclic matrix ex-

pansions uniquely exploit the algebraic structure of cyclic codes to simplify decoding complexity while preserving positional robustness. In addition, combining Appendix H, we provide an explanation and comparison showing that our proposed method does not significantly increase inference latency. More importantly, the performance improvements are not achieved by trading off computational complexity.

More intuitively, Figure 5 provides visual support of our conjecture. As demonstrated in Figures 5a and 5b, the decoder consistently localizes attention to error positions at Layer 2, regardless of where errors occur in the codeword. This uniformity is consistent with the idea that our method unifies error correction strategies under cyclic equivalence. In contrast, Figure 5c shows dispersed attention patterns in Layer 2, indicating multiple competing correction hypotheses that degrade decoding performance. We have more comparisons on the impact of circulant PCM in Appendix F and the entire attention matrices are given in Appendix G. This potentially encourages a research for Transformer-based decoding: intentionally reducing the diversity of error

correction patterns in codes to enhance model focus, thereby achieving performance improvements or being used as a novel incentive in code searching. Such code-aware architectural optimizations align neural decoders more closely with the algebraic properties of target codes, moving beyond generic attention mechanisms toward mathematically grounded decoding strategies.

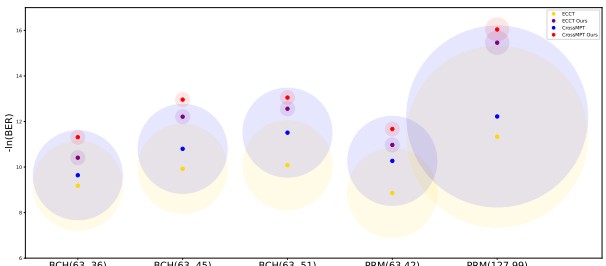

*Figure 6.* A comparison of the number of parameters in different codes by using cyclically parameter reuse in the whole decode layer including feed forward network. A larger area of the circle represents a greater number of parameters.

### 4.2. Parameter Reuse Analysis

This work offers an alternative perspective on the meaning of parameters in Transformer-based decoders than before, using circulant PCM as an entry point to interpret these parameters as encoders of inter-node relationships or their algebraic connections. As empirically validated in Figure 4c, the cyclic reuse of parameter and embedding matrices alongside our scaling strategy achieves a comparable BER to experiments using full $n \times n$ circulant (square) PCMs with unconstrained parameters and embeddings. Additional validation experiments, including both confirmatory and adversarial tests, are provided in Appendix D, while further analyses on the advantages of embedding are presented in Appendix E. These findings might help advance our understanding of decoder operational logic, moving from empirical utility ("how to use") toward mechanistic comprehension ("why it works").

### 4.3. Parameters Decreasing

By reusing parameter and embedding matrices, the model naturally achieves drastic parameter reductions. As shown in Figure 6, our method significantly reduces parameter counts across various decoders and code families while improving performance, using as little as 1.91% (minimum), up to 4.16% (maximum), and averaging less than 3% of the original parameters. We provide additional comparisons in Appendix I. Such dramatic parameter compression makes actual deployment more feasible.

## 5. Conclusion

We propose a novel plug-and-play optimization method for cyclic codes that can be readily deployed across ECCT, CrossMPT, and MM-ECCT decoders, leading to significant reductions in both BER and parameter counts. Using this method as an investigative tool, we empirically analyze the roles of parameters and embedding vectors in Transformer-based decoding and provide empirical support for our interpretations, while demonstrating how the proposed notions—error-correction patterns and inter-node relationships—affect decoding performance. Our aim is to encourage future research that bridges algebraic coding theory and neural decoding architectures, thereby enabling new avenues for code-aware decoder optimizations.

## Acknowledgements

The authors would like to thank the anonymous reviewers for their constructive comments, and Prof. Chong Shangguan for providing access to computational resources during the author-response period, which supported additional experiments and revisions. This research is supported in part by the National Key Research and Development Program of China under Grant Nos. 2022YFA1004900 and 2021YFA1001000, the National Natural Science Foundation of China under Grant No. 62571301.

## Impact Statement

This paper presents work whose goal is to advance the field of machine learning. There are many potential societal consequences of our work, none of which we feel must be specifically highlighted here.

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

# A. Reuse Function and Scaling

In this section, we make the definition of the reuse/reconstruction function $\mathrm{sc}(\cdot)$ precise and explain how it realizes the scaling strategy described in Section 3.3. We first discuss the attention parameter matrices $W_Q, W_K, W_V$ and then turn to the embedding matrix $\hat{W}$.

For better understanding, we illustrate this with the following example:

- When $d = 4n$ ($r = 2$), the attention parameter matrix has size $4n \times 4n$. Compared with $d = 2n$, each scalar relationship between two nodes is now represented by a $2 \times 2$ block. Equivalently, each original node is refined into two virtual sub-nodes, and we need two representative rows for variable nodes and two for check nodes. As we move along the code index, these representative rows are repeated and cyclically shifted with stride $r = 2$, so that the $4n$ rows of the matrix are partitioned into $n$ groups of four rows (two variable, two check) that share the same local pattern up to cyclic shifts. Note that, in this case, the first $2n$ columns correspond to the variable part and the last $2n$ columns to the check part, with cyclic shifts applied independently within each part.

- When $d = n$ ($r = \frac{1}{2}$), the attention parameter matrix has size $n \times n$. In this case, pairs of original nodes share the same parameter row, representing a coarser aggregation of relationships. We still maintain one representative row for variable nodes and one for check nodes, but now there are only $rn = \frac{n}{2}$ aggregated positions conceptually for each of the variable and check parts (both in row and column). As the aggregated index $0, \ldots, rn - 1$ increases, we cyclically shift the representative rows by one position at each step, so that the resulting $n \times n$ matrix still respects the cyclic structure of the code, but at a lower resolution.

More generally, Algorithm 1 provides the definition of the block-wise cyclic reconstruction of the attention parameter matrices for any other admissible value of $r$. The scaling of the embedding matrix $\hat{W} \in \mathbb{R}^{2n \times 2rn}$ is slightly different. As

---

**Algorithm 1** Shift Cyclically (SC)

**Input W′**        /* $\mathbf{W}' \in \mathbb{R}^{2r' \times 2rn}$, $r' = 1$, if $r < 1$; $r' = r$, otherwise. */
**Output W**        /* $W \in \mathbb{R}^{2rn \times 2rn}$ */
Initialize $\mathbf{W} = \mathrm{zeros}(2rn, 2rn)$
/* Fill $\mathbf{W}$ by $\mathbf{W}'$ */
**if** $r >= 1$ **then**
  **for** $ii$ in range(n)  **do**
    /* variable nodes */
    **for** $jj$ in range(r) **do**
      /* $roll(x, s)$ cyclic shifts a vector $x$ by $s$ positions*/
      $\mathbf{W}[r \times ii + jj, : rn] = roll(\mathbf{W}'[jj, : rn], r \times ii)$
      $\mathbf{W}[r \times ii + jj, rn :] = roll(\mathbf{W}'[jj, rn :], r \times ii)$
    /* check nodes */
    **for** $kk$ in range(r) **do**
      /* $roll(x, s)$ cyclic shifts a vector $x$ by $s$ positions*/
      $\mathbf{W}[rn + r \times ii + kk, : rn] = roll(\mathbf{W}'[r + kk, : rn], r \times ii)$
      $\mathbf{W}[rn + r \times ii + kk, rn :] = roll(\mathbf{W}'[r + kk, rn :], r \times ii)$
**else**
  **for** $ii$ in range(rn) **do**
    /* variable nodes */
    $\mathbf{W}[ii, : rn] = roll(\mathbf{W}'[0, : rn], ii)$
    $\mathbf{W}[ii, rn :] = roll(\mathbf{W}'[0, rn :], ii)$
    /* check nodes */
    $\mathbf{W}[rn + ii, : rn] = roll(\mathbf{W}'[1, : rn], ii)$
    $\mathbf{W}[rn + ii, rn :] = roll(\mathbf{W}'[1, rn :], ii)$
**Return W**

---

defined in Section 3.2, the first dimension of $\hat{W}$ is always fixed to $2n$, since each row corresponds to one node (variable or check), only the second dimension is scaled according to $d = 2rn$. We again enforce cyclic-shift invariance along the node index, but now the number of rows is fixed and the cyclic structure is realized purely along the embedding dimension. Then, we always use two representative rows for the embedding matrix: one for variable nodes and one for check nodes. The following Algorithm 2 gives the exact construction. The crucial point is that here we scale only in the second dimension, while the attention parameter matrices are scaled in both dimensions simultaneously.

**Algorithm 2** Shift Cyclically in $\hat{W}$

---

**Input** $\mathbf{W}'$      /* $\mathbf{W}' \in \mathbb{R}^{2 \times 2rn}$ */
**Output** $\hat{W}$      /* $\hat{W} \in \mathbb{R}^{2n \times 2rn}$ */
Initialize $\mathbf{W}$ = zeros($2n, 2rn$)
**for** $ii$ in range(n) **do**
  **if** $r > 1$ **then**
    /* variable nodes */
    $\hat{W}[ii, : rn] = roll(\mathbf{W}'[0, : rn], r \times ii)$
    $\hat{W}[ii, rn :] = roll(\mathbf{W}'[0, rn :], r \times ii)$
    /* check nodes */
    $\hat{W}[n + ii, : rn] = roll(\mathbf{W}'[1, : rn], r \times ii)$
    $\hat{W}[n + ii, rn :] = roll(\mathbf{W}'[1, rn :], r \times ii)$
  **else**
    /* variable nodes */
    $\hat{W}[ii, : rn] = roll(\mathbf{W}'[0, : rn], ii)$
    $\hat{W}[ii, rn :] = roll(\mathbf{W}'[0, rn :], ii)$
    /* check nodes */
    $\hat{W}[n + ii, : rn] = roll(\mathbf{W}'[1, : rn], ii)$
    $\hat{W}[n + ii, rn :] = roll(\mathbf{W}'[1, rn :], ii)$
**Return** $\hat{W}$

---

## B. Reconstruction Method in FFN

In a standard Transformer decoder layer, the position-wise feed-forward network (FFN) consists of two linear layers, $FFN_1 \in \mathbb{R}^{d \times d_{\text{ff}}}$, $FFN_2 \in \mathbb{R}^{d_{\text{ff}} \times d}$, where $d$ is the embedding dimension and $d_{\text{ff}}$ is the hidden dimension (typically $d_{\text{ff}} = 4d$). In our setting, the embedding dimension of each layer is scaled as $d = 2rn$, so that the hidden dimension becomes $d_{\text{ff}} = 4d = 8rn$. We can understand $FFN_1$ using the same scaling strategy as for the attention parameter matrices: along the first dimension (rows), the scaling is identical to that of the attention parameter matrices, while along the second dimension (columns) the length is four times larger, so the number of shifts is expanded by a factor of four. Accordingly, we introduce Algorithm 3 to describe the reconstruction of $FFN_1$. For $FFN_2$, we simply treat its transpose as an instance of $FFN_1$ and apply the transposed version of the Algorithm 3.

---

**Algorithm 3** Shift Cyclically in FFN

---

**Input** $\mathbf{W}'$      /* $\mathbf{W}' \in \mathbb{R}^{2r' \times 8rn}$, $r' = 1$, if $r < 1$; $r' = r$, otherwise. */
**Output** $\mathbf{W}$      /* $\mathbf{W} \in \mathbb{R}^{2rn \times 8rn}$ */
Initialize $\mathbf{W}$ = zeros($2rn, 8rn$)
/* Fill $\mathbf{W}$ by $\mathbf{W}'$ */
**if** $r >= 1$ **then**
  **for** $ii$ in range(n) **do**
    /* variable nodes */
    **for** $jj$ in range(r) **do**
      /* $roll(x, s)$ cyclic shifts a vector $x$ by $s$ positions*/
      $\mathbf{W}[r \times ii + jj, : 4rn] = roll(\mathbf{W}'[jj, : 4rn], 4r \times ii)$
      $\mathbf{W}[r \times ii + jj, 4rn :] = roll(\mathbf{W}'[jj, 4rn :], 4r \times ii)$
    /* check nodes */
    **for** $kk$ in range(r) **do**
      /* $roll(x, s)$ cyclic shifts a vector $x$ by $s$ positions*/
      $\mathbf{W}[rn + r \times ii + kk, : 4rn] = roll(\mathbf{W}'[r + kk, : 4rn], 4r \times ii)$
      $\mathbf{W}[rn + r \times ii + kk, 4rn :] = roll(\mathbf{W}'[r + kk, 4rn :], 4r \times ii)$
**else**
  **for** $ii$ in range(rn) **do**
    /* variable nodes */
    $\mathbf{W}[ii, : 4rn] = roll(\mathbf{W}'[0, : 4rn], 4ii)$
    $\mathbf{W}[ii, 4rn :] = roll(\mathbf{W}'[0, 4rn :], 4ii)$
    /* check nodes */
    $\mathbf{W}[rn + ii, : 4rn] = roll(\mathbf{W}'[1, : 4rn], 4ii)$
    $\mathbf{W}[rn + ii, 4rn :] = roll(\mathbf{W}'[1, 4rn :], 4ii)$
**Return** $\mathbf{W}$

---

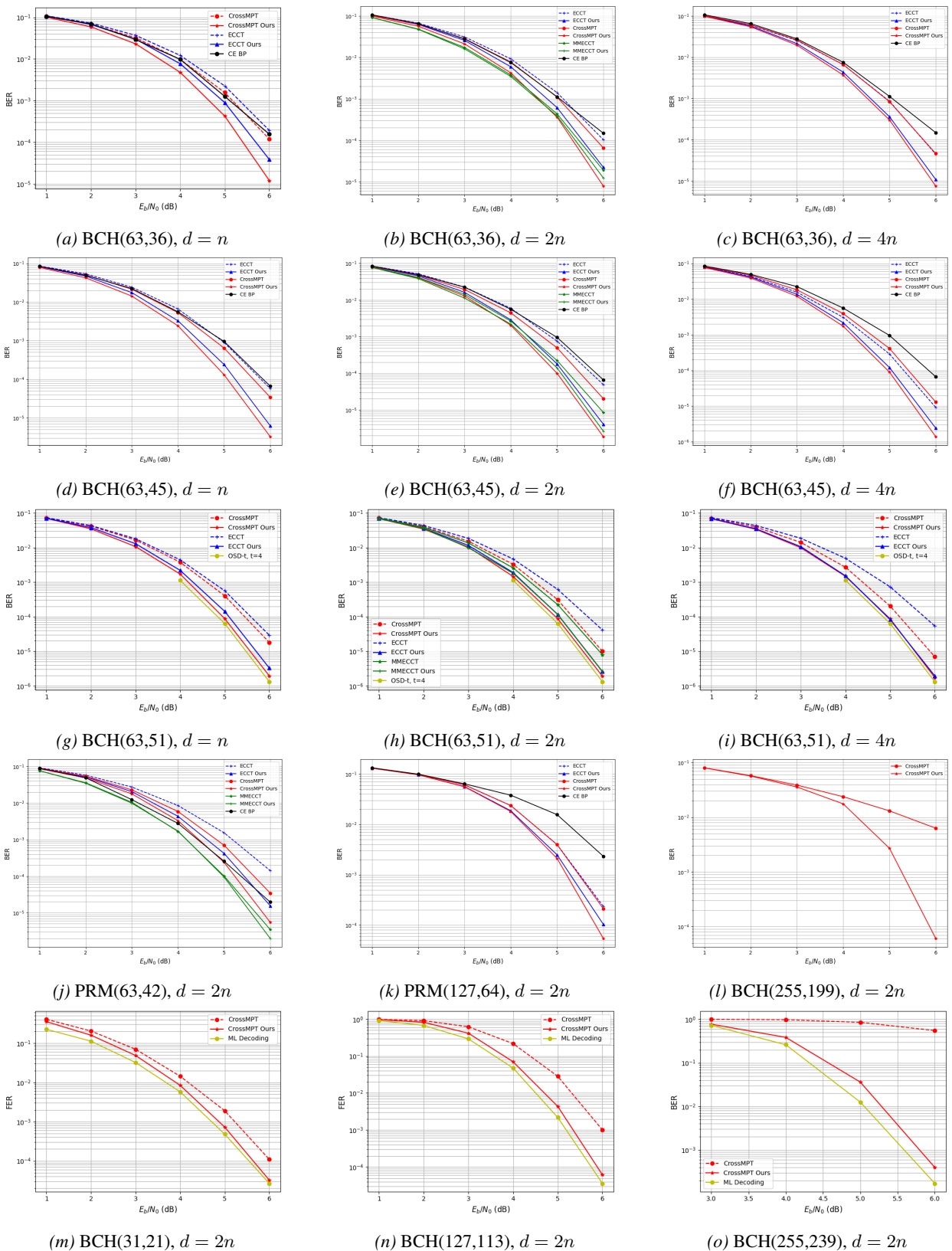

*(a)* BCH(63,36), $d = n$

*(b)* BCH(63,36), $d = 2n$

*(c)* BCH(63,36), $d = 4n$

*(d)* BCH(63,45), $d = n$

*(e)* BCH(63,45), $d = 2n$

*(f)* BCH(63,45), $d = 4n$

*(g)* BCH(63,51), $d = n$

*(h)* BCH(63,51), $d = 2n$

*(i)* BCH(63,51), $d = 4n$

*(j)* PRM(63,42), $d = 2n$

*(k)* PRM(127,64), $d = 2n$

*(l)* BCH(255,199), $d = 2n$

*(m)* BCH(31,21), $d = 2n$

*(n)* BCH(127,113), $d = 2n$

*(o)* BCH(255,239), $d = 2n$

*Figure 7.* The BER, FER performance of our method used in ECCT, CrossMPT, MM-ECCT.

## C. More Plots of Performance

As illustrated in Figure 7, we present an extensive set of plots to give a more complete picture of the performance gains achieved by our method. In BER comparison, across all cases that we had considered, our approach delivers improvements that span one to several orders of magnitude, and in certain situations, the curve even approaches that of the Ordered Statistics Decoder (OSD), highlighting the broad applicability of the proposed gains. In FER comparison, at high code rates (i.e., $\frac{k}{n}$), our method not only leaves the original decoder far behind but also comes close to the maximum likelihood (ML) decoding performance. Due to the comparison of FER, the results provide compelling evidence that, in realistic deployment settings, the proposed method offers effective and efficient performance enhancements.

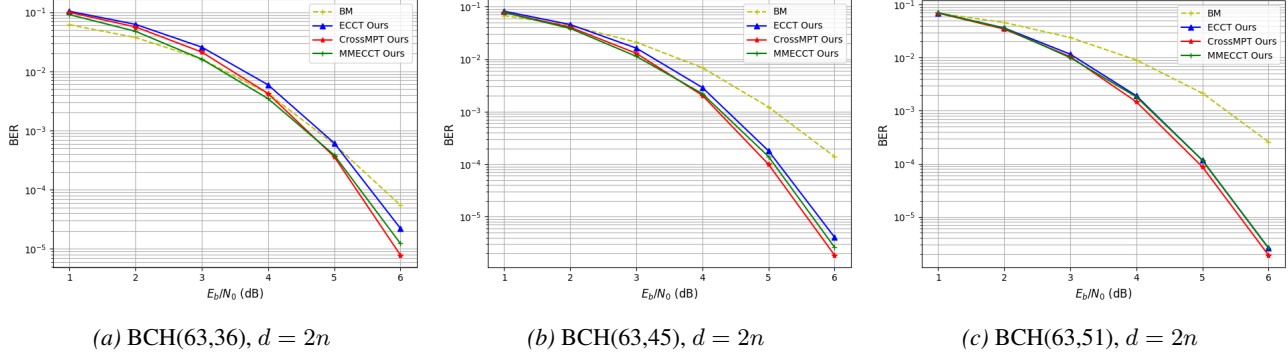

*(a)* BCH(63,36), $d = 2n$      *(b)* BCH(63,45), $d = 2n$      *(c)* BCH(63,51), $d = 2n$

*Figure 8.* The BER performance compared with Berlekamp–Massey algorithm.

The Berlekamp–Massey algorithm is the well-known traditional decoding method for BCH codes. As shown in Figure 8, we compare the Berlekamp–Massey decoder with ECCT, CrossMPT, and MM-ECCT after being enhanced by our approach. It can be observed that the improved models consistently outperform the traditional algorithm by one to two orders of magnitude.

## D. More Comparison in Embedding

*Table 3.* We compare our parameter-sharing scheme with a fully trainable matrix scheme in ECCT, CrossMPT, and MM-ECCT, both using the same $n \times n$ circulant PCM.

| Decoder | ECCT | | | CrossMPT | | | MM-ECCT | | | ECCT Ours | | | CrossMPT Ours | | | MM-ECCT Ours | | |
|---|---|---|---|---|---|---|---|---|---|---|---|---|---|---|---|---|---|---|
| Code/SNR | 4 | 5 | 6 | 4 | 5 | 6 | 4 | 5 | 6 | 4 | 5 | 6 | 4 | 5 | 6 | 4 | 5 | 6 |
| BCH(63,36) | 5.00 | 7.19 | 10.27 | 5.44 | 7.92 | 11.63 | 5.41 | 7.79 | 11.29 | **5.13** | **7.40** | **10.71** | 5.47 | 7.93 | 11.75 | 5.65 | 7.79 | 11.31 |
| BCH(63,45) | 5.72 | 8.23 | 11.96 | 6.19 | 9.21 | 13.16 | 6.12 | 8.88 | 12.84 | **5.85** | **8.63** | **12.41** | 6.19 | 9.21 | 13.19 | 6.12 | 8.88 | 12.85 |
| BCH(63,51) | **6.27** | 9.06 | **12.88** | **6.53** | 9.35 | 13.11 | 6.31 | **9.06** | 12.83 | 6.26 | **9.06** | 12.86 | 6.52 | **9.35** | 13.17 | 6.31 | 9.05 | **12.86** |

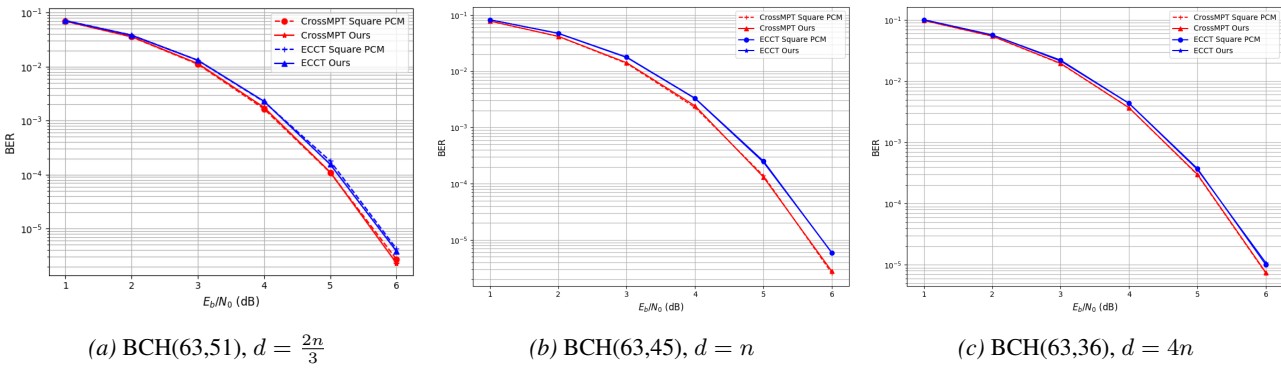

*(a)* BCH(63,51), $d = \frac{2n}{3}$      *(b)* BCH(63,45), $d = n$      *(c)* BCH(63,36), $d = 4n$

*Figure 9.* Comparing the variation of using parameter reuse with both use square PCM in decoders (a) (b) (c).

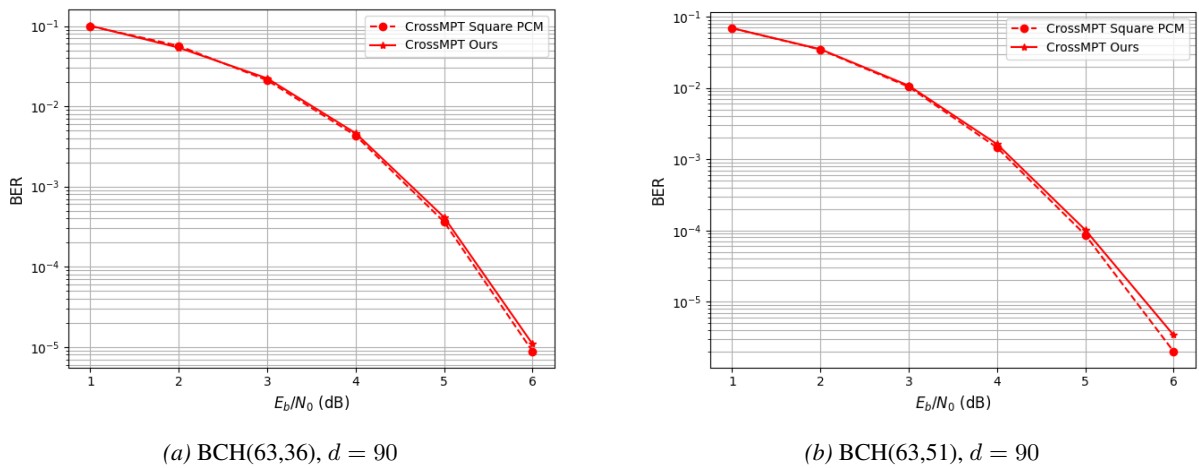

*(a)* BCH(63,36), $d = 90$          *(b)* BCH(63,51), $d = 90$

*Figure 10.* Comparing the variation of using parameter reuse that did not comply with the scaling by setting $d = 90$ (a) (b).

To better verify the correctness of our method, as shown in Table 3, when using the same $n \times n$ circulant parity-check matrix, our approach achieves performance comparable to or in some cases slightly better than the original scheme that trains all parameters.

When properly applying our scaling strategy, as shown in Figures 9, our method achieves performance metrics identical to the full-parameter baseline for BCH(63,51) with $d = \frac{2n}{3}$, BCH(63,45) with $d = n$ and BCH(63,36) with $d = 4n$. In contrast, Figure 10 demonstrates that incorrect scaling (e.g. $d = 90$) introduces mismatches in the cyclic reuse of parameter and embedding matrices, leading to measurable performance degradation. These counterexamples indirectly support the correctness of our theoretical interpretations regarding parameter and embedding interactions.

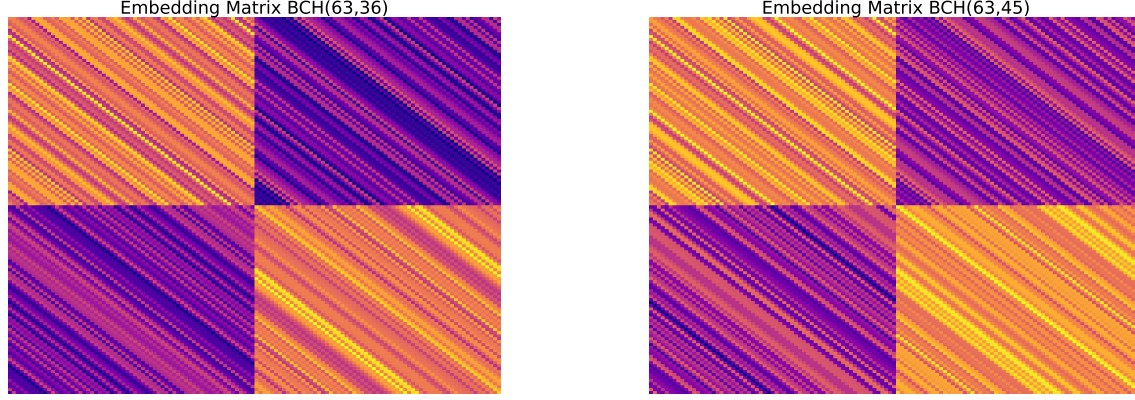

*Figure 11.* The heatmap of the embedding matrix $\hat{W}$ reconstructed by Algorithm 2

As shown in Figure 11, we present the heatmap of the embedding matrix reconstructed by Algorithm 2 after training. We observe that our method naturally partitions the embedding matrix into four regions, each of which directly corresponds to one type of inter-node relationship. This, in turn, provides indirect evidence supporting our interpretation that the embedding vectors encode relations between nodes.

## E. Balance of Performance and Complexity

With a larger embedding dimension $d$, the model can capture finer-grained features of each code feature that we verify in this paper can be interpreted as inter-codes relationships, to reduce the BER. However, the price paid is a growth in computational

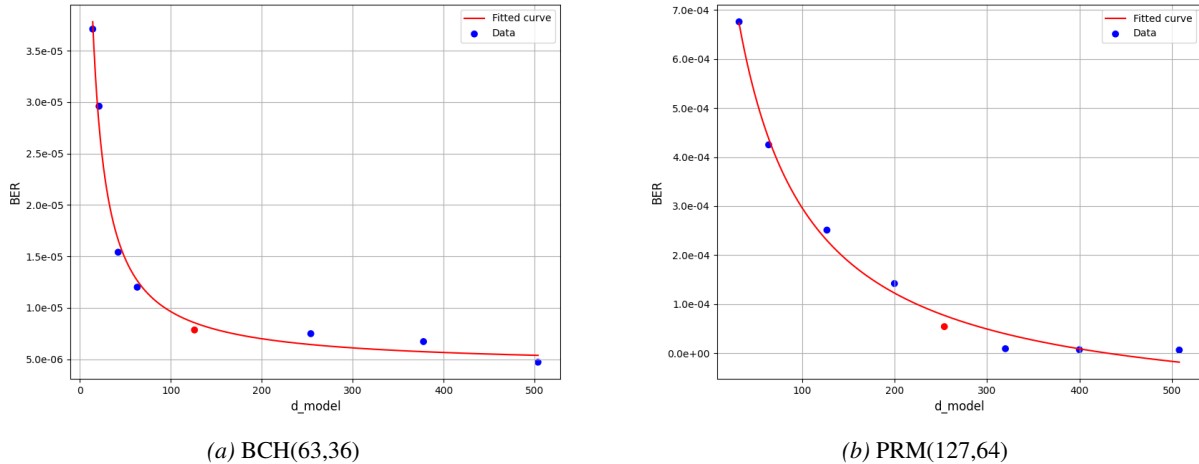

*(a)* BCH(63,36)  *(b)* PRM(127,64)

*Figure 12.* Scatter plot and fitted curve illustrating how the BER varies with embedding dimension.

and space complexity that soon becomes unacceptable. As shown in Figure 12, the BER falls roughly in inverse proportion to $d$, implying that infinitely increasing the embedding dimension to further chase lower BER is unjustified. When $d < 2n$, the model does not yet express inter-codes relationships adequately, so expanding the embedding dimension brings clear performance gains. However, when $d > 2n$, these relationships appear to have been largely captured; any further increase in $d$ merely provides a more detailed but only marginally beneficial representation in our experiments. The critical point occurs at $d = 2n$, where the embedding dimension matches the length of the node count and forms a one-to-one correspondence with the code bits. This value (i.e., $d = 2n$) emerges as the inflection point of the fitted curve, marking the near-optimal trade-off between performance and complexity.

*Table 4.* Following the convention in ECCT (Choukroun & Wolf, 2022), we compares the sparsity ratio of the attention matrices (higher is better) and the computational complexity ratio (lower is better). For MM-ECCT, the reported value corresponds to the mask matrix constructed from a systematic parity-check matrix, so it can be compared directly with ECCT. In CrossMPT, however, the mask is taken directly from the original $(n - k) \times n$ PCM (smaller than the mask used in ECCT/MM-ECCT), and thus its sparsity and complexity are not directly comparable with those of ECCT and MM-ECCT.

| Decoder | ECCT | | MM ECCT | | ECCT Ours | | CrossMPT | | CrossMPT Ours | |
|---|---|---|---|---|---|---|---|---|---|---|
| Code | Sparsity | Complexity | Sparsity | Complexity | Sparsity | Complexity | Sparsity | Complexity | Sparsity | Complexity |
| BCH(63,36) | 51.48% | 24.26% | 58.22% | 20.89% | **60.32%** | **19.84%** | 71.43% | 28.57% | 71.43% | 28.57% |
| BCH(63,45) | 36.12% | 31.94% | 46.91% | 26.54% | **55.56%** | **22.22%** | 61.9% | 38.1% | 61.9% | 38.1% |
| BCH(63,51) | 26.35% | 36.83% | 30.15% | 34.92% | **52.38%** | **23.81%** | 55.56% | 44.44% | 55.56% | 44.44% |
| PRM(127,99) | 40.1% | 29.95% | 40.34% | 29.83% | **62.99%** | **18.5%** | 74.8% | 25.2% | 74.8% | 25.2% |

As shown in Table 4, we compare the changes in sparsity ratio and computational complexity ratio before and after applying our method. The results demonstrate that, after incorporating our method, the modified ECCT and MM-ECCT achieve the best overall figures. In addition, because our approach cyclically extends the parity-check matrix (with each added row having the same weight), the resulting sparsity ratio and computational complexity ratio remain unchanged relative to CrossMPT.

As shown in Table 5, our scheme does indeed incur an increase in computational cost in some cases. However, we believe that this additional cost is well justified. We provide a detailed explanation below.

- While preserving the original Transformer decoding framework, i.e., maintaining plug-and-play compatibility, the three key metrics of performance, computational complexity, and memory consumption can be viewed as a triangular trade-off that is difficult to optimize simultaneously. Although our method does introduce some additional computational complexity, it significantly improves decoding performance while substantially reducing memory usage. For example, Table 1 and Figure 5 have already demonstrated that our scheme achieves a large performance gain over the original model. In Figure 7(o), for the longer BCH(255, 239) code, our method even achieves a performance improvement

*Table 5.* Since our method adopts a circulant matrix as PCM, it may potentially increase the overall computational complexity. Therefore, in the following table, we provide a comparison of FLOPs, memory consumption, and performance.

| Decoder | CrossMPT | | | CrossMPT Ours | | |
|---|---|---|---|---|---|---|
| Code | FLOPs(M) | Memory(G) | -ln(BER) | FLOPs(M) | Memory(G) | -ln(BER) |
| BCH(63,36) | 216.21 | 2.12 | 9.64 | 282.30 | 1.18 | 11.75 |
| BCH(63,45) | 192.19 | 1.51 | 10.80 | 282.30 | 1.18 | 13.19 |
| PRM(127,64) | 1863.42 | 4.67 | 8.47 | 2257.34 | 3.52 | 9.83 |
| PRM(127,99) | 1683.94 | 3.95 | 12.22 | 2257.34 | 3.52 | 16.61 |
| PRM(511,256) d=73 | 1045.96 | 13.5 | 4.04 | 1251.31 | 6.72 | 5.55 |
| PRM(1023,638) d=31 | 784.39 | 25.76 | 4.36 | 1285.44 | 18.7 | 5.86 |

of nearly three orders of magnitude. Meanwhile, as shown in Table 5, our scheme also greatly reduces memory consumption. For PRM(1023, 638), the original model cannot even run on an RTX 4090 due to excessive memory usage, in which case memory becomes the practical bottleneck. Therefore, within this triangular trade-off, although we sacrifice a certain amount of computational complexity, the resulting gains are worthwhile.

- The performance improvement should not be understood simply as a consequence of increased computation. As shown in Table 2, even under the same computational budget, our scheme still maintains a substantial performance advantage. This indicates that the performance gain is primarily attributable to our architectural design rather than merely to the increase in computational cost.

- The impact of the increased computation is not severe in practice. As reflected by the decoding latency in Table 5, due to the parallel computing capability of GPUs, the actual latency is not linearly proportional to the absolute number of floating-point operations. Therefore, although the additional overhead is non-negligible, the practical increase in latency is much milder than what might be expected from the theoretical FLOPs alone.

# F. Impact of Training convergence

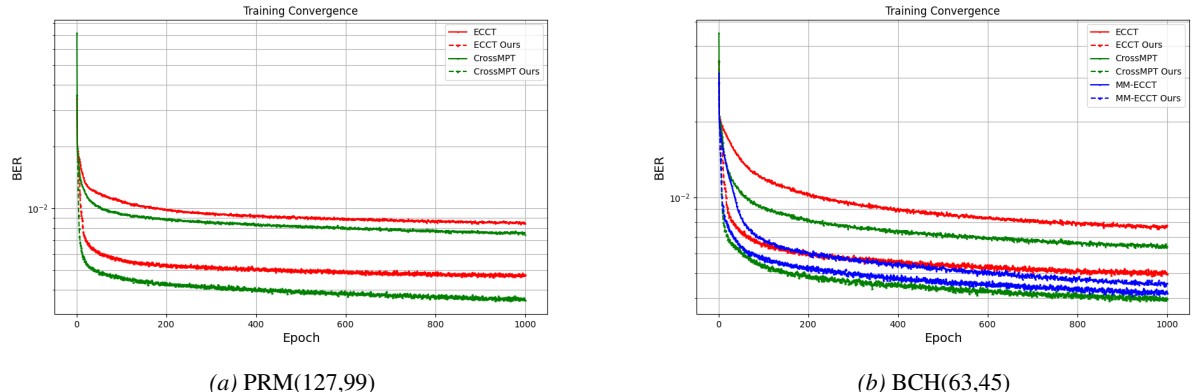

*(a)* PRM(127,99)  *(b)* BCH(63,45)

*Figure 13.* Comparing the impact of training convergence within our method.

As is well known, the M constructed from the PCM can effectively improve the communication efficiency between code bits. Therefore, the improved design of the PCM is of great interest. To validate the effectiveness of our method, we observe the change in BER during each epoch. As shown in Figure 13, it can be seen that our method leads to a faster decrease in BER compared to the original decoder, with the same amount of training. This indicates that even with fewer training epochs, our method can still maintain good performance.

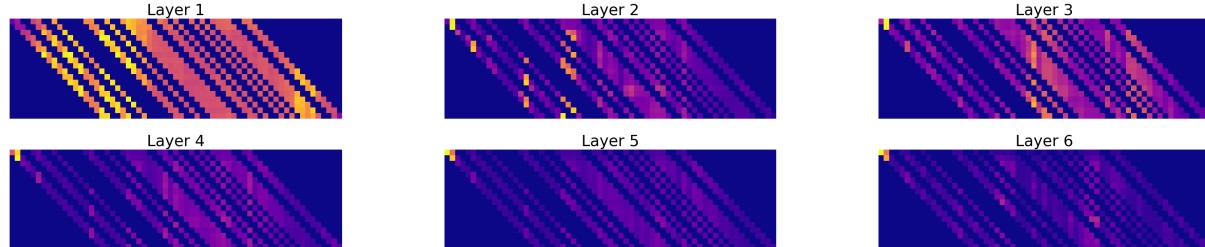

*Figure 14.* The entire attention matrices in original CrossMPT's decode layer with a single error bit at 2nd position, when $N = 6$.

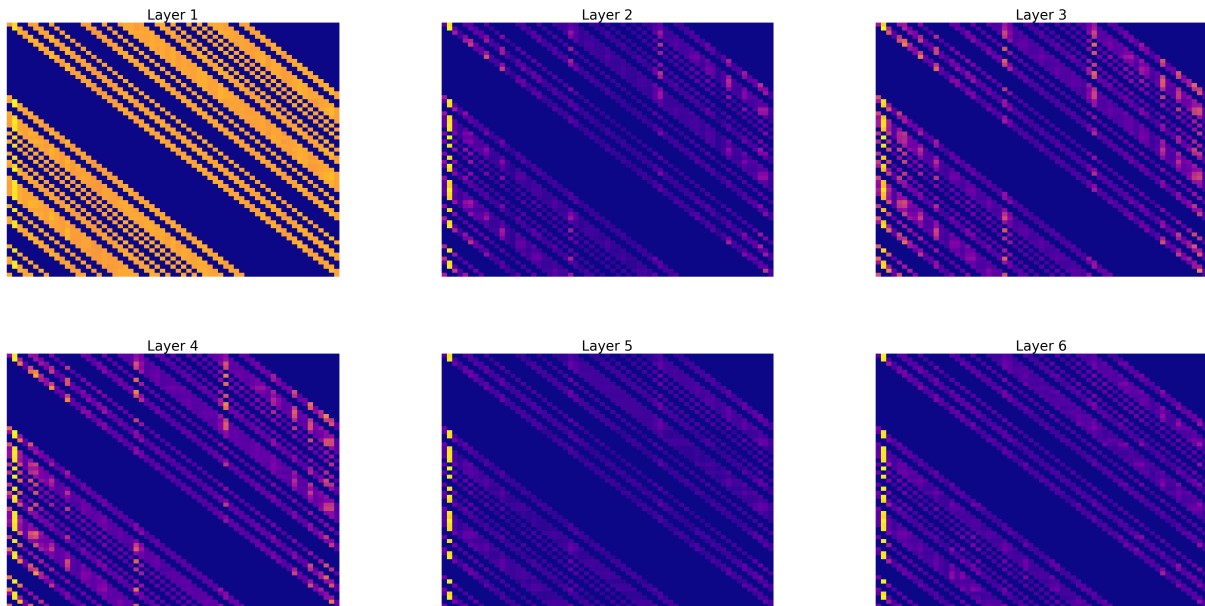

*Figure 15.* The entire attention matrices in decode layer with a single error bit at 2nd position by using our method, when $N = 6$.

## G. Entire Attention Score

The attention matrix typically reflects the areas that the model is focusing on. In a decoder, it should indicate the areas where the model suspects errors may occur. To provide a more comprehensive reflection of the error correction capability of our method, as shown in Figures 14 and 15, we list the attention matrices of all decode layers while decoding BCH(63,45). We can intuitively observe that our method primarily focuses on the error positions (i.e., the second column) throughout the entire global region. However, in the original CrossMPT, the first three layers focus on many other positions, and even after convergence in the later layers, they do not primarily focus on the second position but instead shift their attention to the first position (the first column).

This suggests that our method is able to quickly identify the error position during decoding and maintain attention on them, thereby reducing the BER. At the same time, it also validates our explanation of the error correction pattern, which suggests that there is only one cyclically shiftable error correction pattern (i.e., once the error position is identified, the model will focus solely on that area).

## H. Inference Time

Admittedly, our scheme increases the input sequence length. However, the comparisons in Tables 2 and 4 indicate that the observed performance gains are not achieved by trading off increased computational complexity. Instead, both the sparsity ratio and the computational complexity ratio decrease, reaching state-of-the-art values. Consequently, as shown in Table 6,

*Table 6.* The inference time compared with CrossMPT (per codeword).

| Decoder/Code | CrossMPT | CrossMPT Ours |
|---|---|---|
| BCH(63,36) | 202.93 $\mu$s | 236.685 $\mu$s |
| BCH(63,45) | 183.325 $\mu$s | 237.025 $\mu$s |
| PRM(127,64) | 370.968 $\mu$s | 419.136 $\mu$s |
| PRM(127,99) | 335.515 $\mu$s | 419.469 $\mu$s |

compared with the original CrossMPT, our scheme does not lead to a significant increase in inference time.

## I. The Number of Parameters

In Figure 16, we employ histograms to provide a direct visual comparison of the drastic reductions in parameter counts achieved by our method.

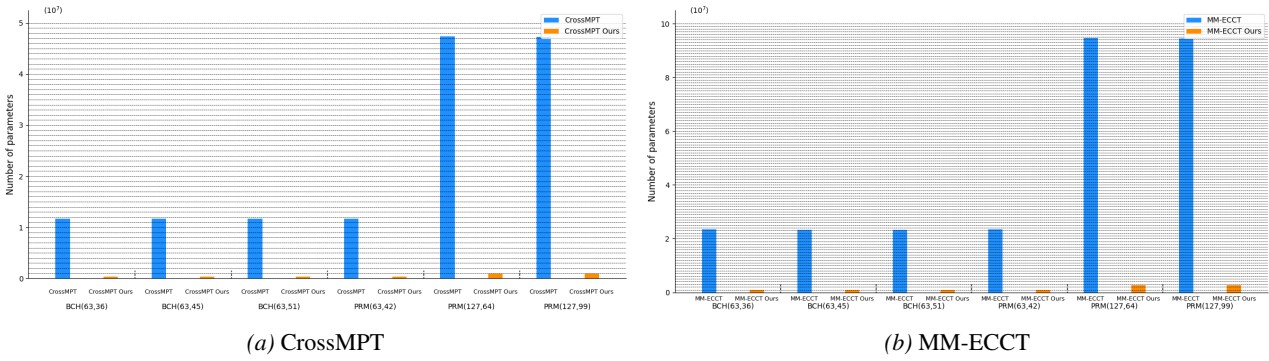

*(a)* CrossMPT  *(b)* MM-ECCT

*Figure 16.* A comparison of the number of parameters between used our method and traditional method in, CrossMPT, MM-ECCT

## J. The Use of LLMs

All the ideas, experiments, figures, and writing details in this paper were completed by the authors; LLMs were involved only in the final stage for polishing and grammatical corrections.

