# OpenReview forum: "Drop-in Circulant Structural Priors for Transformer Decoding of Cyclic Codes"
_ICML.cc/2026/Conference — ICML 2026 regular_

### Official Review · Reviewer_VTme · 2026-03-05

**Soundness:** 3
**Presentation:** 3
**Significance:** 2
**Originality:** 3
**Overall Recommendation:** 5
**Confidence:** 3

**Summary:**

This work addresses the optimization of Transformer-based neural decoders for error correction codes by exploiting code-specific algebraic structure rather than treating codes as generic sequences. This manuscript discuss whether the inherent cyclic structure of certain code families (BCH, Punctured Reed-Muller) can be systematically embedded into neural architectures to simultaneously improve decoding performance and reduce model complexity. The authors demonstrate that under cyclic equivalence the learning task/weights count can be drastically simplified. The proposed method is designed as a plug-and-play module compatible with existing architectures (ECCT, CrossMPT, MM-ECCT), requiring no architectural redesign. Empirically, the authors report significant improvement in bit error rate (BER) while reducing the total parameter count.

**Compliance With Llm Reviewing Policy:**

Affirmed.

**Final Justification:**

This paper proposes a plug-and-play cyclic structure exploitation method for Transformer-based neural decoders, demonstrating meaningful BER improvements alongside significant parameter reduction. The core ideas are sound and the empirical validation is comprehensive.

My initial concerns centered on:
- Notation ambiguity in the PCM representation and unconventional use of negative log-BER in tables.
- Fairness of baseline comparisons and the lack of comparison with conventional decoders in terms of complexity.

The authors' rebuttal satisfactorily addressed my main concerns. The acknowledgment that the method does not aim to outperform standard decoders in raw efficiency, but rather to improve Transformer-based decoders, is a reasonable framing, although improved comparison with said decoders would have strengthened the content. The additional results and the responses to other reviewers further strengthen the submission. Remaining minor concerns (e.g. applicability for QC-LDPC codes) are acknowledged by the authors as future work directions, which I find acceptable.

Overall, the rebuttal has addressed my main concerns and reinforced my positive assessment. I update my presentation score to "good", reflecting the provided clarifications, and I maintain my recommendation of **"Accept"**.

I thank the authors for their comprehensive responses and the effort put into addressing the reviewers' concerns.

**Key Questions For Authors:**

- 1.	My main concern is related to the PCM matrix used in the experiment both for the proposed technique and the baseline decoder (Table 1 and 2). Do you consider the standard PCM BCH and PRM matrices (which are cyclic codes) or do you expand them to square circulant matrices. If so, do you expand the PCM to $n \times n$ for all decoders (including baselines), or only for your method? If only for your method: Are you comparing at equal $E_b/N_0$ or equal SNR? Please clarify how you ensure fair comparison. I feel this confusion comes from the initial explanation of the circulant expansion of the Hamming (7,4) code which leads to believe that this is a prerequisite of the method to consider only square circulant matrices (and not just cyclic codes).
- 2.	The method is tested only on BCH and PRM codes. What about quasi-cyclic codes (e.g., QC-LDPC)? These are widely used in practice (5G NR) and share partial cyclic structure. Table 2 shows the method fails for POLAR and LDPC codes, but no analysis is provided for why or how the approach could be adapted.
- 3.	While parameter reduction is impressive, how does your method compare to conventional algebraic decoders (e.g. OSD) in terms of FLOPs, hardware complexity, and/or latency? Also, the method requires training, which is not needed for algebraic decoders. What is the training cost vs. deployment benefit trade-off?

**Limitations:**

Yes.

**Strengths And Weaknesses:**

Here is a general appreciation of the paper **strength**. The authors rely on the algebraic structure of codes/decoders to condition the processing and architecture of a class of model-free decoder (hence injecting expert knowledge within the model free aspect). Using the redundant error correction patterns and internode relationships present in cyclic codes, the paper devise a way to share parameters, reducing model complexity while improving performance. One strength of the approach is the agnostcity to the architecture used (at least for the considered family of transformer based decoder). The paper demonstrate interesting performance improvements with significant weight reduction. A comprehensive validation shows the difference with plain ECCT and CrossMPT architectures. The additional appendices are appreciable addition to the paper. The underlying principles of error correcting codes, cyclic structure and transformer based are well explained in introduction.

Now follows a list of paper **weakness** with suggestions of improvements:

- The use of both BER and FER metrics is appropriate for ECC evaluation. I appreciate its use in Figure 4.a. I would appreciate this for all performance Table/Figure as it make the direct comparison with most paper of the litterature much easier. The negative natural logarithm of the BER used in Table 1, appear unconventional for a coding paper and complicates unnecessarily the interpretation of the results. Standard practice in coding theory involves presenting BER or block error rate (BLER/FER) results directly, which allows for clearer results.
- I feel there is a notation ambiguity and concerns about the interpretation of the circulant PCM expansion. In figure 1, the transition from standard $(n-k)\times n$ PCM to $n \times n$ circulant PCM creates confusion about what variables $v_1 \dots v_n$ represent. In the standard Hamming (7,4): $v_1 \dots v_7$ denote the entire codeword (4 info + 3 parity bits), but in the proposed circulant case: It appears $v_1 \dots v_n$ denote only information bits, with $c_1 \dots c_n$ as parity bits, yielding $2n$ total variables (as hinted line 197, col 1). This difference in notation convention would benefit being explicitely stated to avoid confusion. To my understanding, in the Hamming example, the PCM columns represent the entire codeword, while in the circulant case, they represent only the information bits, with parity bits being implicit outputs of the constraints.
- Similar confusion remains in Table 1 and Table 2 as to the expression of the PCM used (both for proposed decoder and classic ones). It is not very clear if the matrix are just cyclic matrix (eg standard BCH) or they circulant square expansion. If an expansion is used it is not clear whether baseline decoders (ECCT, CrossMPT without cyclic reuse) use the same expanded PCM or the original PCM. If expanded PCMs are used only for the proposed method, the comparison may be unfair if not accounting for proper $E_b/N_0$ evaluation. The explanation of expansion method used in Table 2 is not very clear to me either.
- In Figure 3 the "Frozen for decoding" caption is unclear to me.
- No comparison with conventional decoders (e.g., OSD) in terms of computational complexity (FLOPs, latency) is provided, which would probably make sense as the paper look at the question of complexity reduction compared to transformer architectures. The complexity aspect in Table 4 is not clearly explained to me. Table 5 provides inference time but not operation counts which says more about implementation rather than inherent complexity. The CE BP baseline uses only 5 iterations, which may not be complexity-equivalent to the Transformer architecture (even with the proposed complexity reduction). BP is also known to be suboptimal for BCH codes. This should be acknowledged as a limitation of the BP baseline choice.

Now follow a general appreciation regarding all four evaluation items:
- **Soundness:**  The core ideas are sound and well-supported empirically, but notation ambiguities, incomplete baseline descriptions, and lack of formal proofs for key claims could be improved.
- **Presentation:** The paper is generally well-written, but ambiguities (PCM expansion, baseline setup) and missing comparisons significantly hinder clarity.
- **Significance:** The work makes solid contributions to neural ECC decoding, but limited scope (only cyclic codes).
- **Originality:** The plug-and-play design is elegant. However, parameter sharing itself is not new, and differentiation from prior would benefit strengthening.

---

> ### Author Rebuttal · Authors · 2026-03-31
>
> Dear Reviewer,
>
> We would like to express our sincere gratitude for your time and effort in reviewing our manuscript. After carefully considering the comments, we provide our detailed responses to each point below.
>
> **Ambiguity in Notation and Representation**
>
> Regarding the representation format, we sincerely apologize for the current use of the negative natural logarithm in the tables. This choice was influenced by a series of prior related works, but we will revise it in the subsequent version for better clarity. Throughout the paper, columns of the PCM always correspond to codeword coordinates (hence variable nodes), not only to information bits. The $n\times n$ circulant square extension adds redundant parity-check equations (rows), but does not change the set of codeword variables (columns). Thus, in the $(7,4)$ Hamming example, we still have 7 variable nodes corresponding to the 7 codeword positions; what changes is that the number of check-node rows is extended from 3 to 7
>
> **Experimental Setup**
>
> In Table 1, ECCT and CrossMPT baselines use the original PCM constructions from their original papers. Our method modifies the proposed model by replacing the original PCM with the $n\times n$ circulant square expansion. In Table 2, to control for the increased sequence length / compute, we compare against CrossMPT-Rselect, which also uses an $n\times n$ square PCM, but obtained by random expansion instead of cyclic expansion.
>
> **Scalability**
>
> As the reviewer insightfully pointed out, the cyclic-equivalent reuse mechanism in our method can indeed be naturally transferred to the block-circulant structure of QC-LDPC codes. However, such a transfer mainly leads to a substantial reduction in the number of parameters, and does not by itself guarantee a performance improvement. How to extend the parity-check matrix of QC-LDPC codes in a principled and interpretable manner, so as to further translate this structural prior into performance gains, is an important direction for our future work.
>
> In addition, the experiments on POLAR and LDPC codes in Table 2 are intended to show that randomly expanding the PCM into a square matrix does not necessarily lead to performance improvement.
>
> Frozen for decoding means that the full parameter matrices are reconstructed from a small set of learnable representative rows before the iterative decoding steps of a forward pass, and are then used as fixed weights during decoding. During training, gradients are still propagated back to the representative rows.
>
> **Comparison with Traditional Decoders**
>
> The relatively high computational complexity can indeed be regarded as a common limitation of Transformer-based decoders. However, recent advances, such as the quantization-based acceleration proposed by Matan Levy et al. and EfficientMPT proposed by S. J. Park et al., have the potential to substantially alleviate this burden. And we agree that our goal is not to outperform algebraic decoders such as OSD/Berlekamp–Massey in raw efficiency. Rather, our goal is to improve existing Transformer-based decoders. Therefore, compared with traditional decoders, our method does not have a clear advantage in terms of computational complexity. Our comparison with the OSD decoder in the manuscript is intended to show that the performance of our method has already reached a fairly competitive level. In addition, we think that the application scenarios of these two types of decoders may not be the same. For example, NVIDIA has recently begun integrating coding and decoding algorithms into GPUs, which may create new deployment opportunities for neural decoding methods.
>
> **Rayleigh Fading Channel**
>
> To verify the consistency of the performance improvement brought by our method, we further include in the table below a comparative evaluation over the Rayleigh Fading Channel. The results show that our method still maintains a substantial performance advantage.
>
> |Decoder||CrossMPT|||CrossMPT Ours||
> |:-:|:-:|:-:|:-:|:-:|:-:|:-:|
> ||4|5|6|4|5|6|
> |BCH(63,45)|3.49e-2|2.08e-2|1.13e-2|2.43e-2|1.18e-2|5.25e-3|
> |BCH(63,51)|3.63e-2|2.30e-2|1.34e-2|2.79e-2|9.89e-3|6.24e-3|
>
> Finally, we will carefully consider and incorporate your valuable suggestions. We would also like to sincerely thank you again for your recognition of our work and for the time and effort you devoted to this review.

---

> > ### Author Rebuttal · Reviewer_VTme · 2026-04-01
> >
> > All concerns have been fully resolved. The authors' responses have clarified the experimental setup and the PCM representation/construction. I also found the responses to other reviewers informative, particularly the additions regarding longer codes and Memory/FLOPs data. Thank you for the comprehensive rebuttals.

---

### Official Review · Reviewer_yUf8 · 2026-03-11

**Soundness:** 3
**Presentation:** 2
**Significance:** 3
**Originality:** 3
**Overall Recommendation:** 4
**Confidence:** 3

**Summary:**

This paper proposes a circulant-structure-aware Transformer decoder for cyclic error-correcting codes. The core idea is that, instead of feeding a standard parity-check matrix into a generic Transformer, the decoder uses a square circulant parity-check matrix and cyclic parameter sharing so the model explicitly matches the symmetry of cyclic codes. The authors argue that this makes all bit positions follow equivalent error-correction patterns, which simplifies what the network must learn; empirically, the method improves BER on BCH and punctured Reed–Muller codes, often by about an order of magnitude at high SNR, while also reducing the number of trainable parameters to only a few percent of the original model. Overall, the paper’s main contribution is showing that injecting algebraic code structure directly into Transformer architecture and parameterization can yield better decoding performance and much more parameter-efficient models.

**Compliance With Llm Reviewing Policy:**

Affirmed.

**Final Justification:**

The rebuttal cleared my concerns and comments. Based on the paper's contents and the rebuttal, I think the paper has a clear contribution and advantages. So, I am leaning toward acceptance.

**Key Questions For Authors:**

Please take a look at the weaknesses above. Specifically, I would ask for one tighter theorem or formal proposition with proof in the main paper, not just appendicial reconstruction algorithms; one stronger disentangling ablation separating circulant-PCM gains from reuse gains; and one broader generalization experiment or a stricter equal-compute comparison.

**Strengths And Weaknesses:**

## Strengths
This paper studies the question whether neural decoders should remain generic sequence models, or whether they should explicitly inherit algebraic structure from the target code family. Specifically, the paper investigates whether the benefits of larger embeddings and many Transformer parameters can be reinterpreted in structural terms, rather than treated as opaque capacity. On both questions, the paper offers a coherent, interesting answer: for cyclic codes, use a circulant PCM and enforce cyclic parameter reuse. That answer is technically clean, easy to plug into existing decoders, and empirically strong. A positive contribution of the paper is that the work is not just “one more decoder tweak.” It gives a mechanistic story for why the tweak should work, and the ablations are reasonably aligned with their claim: random square expansion helps less than circulant expansion; reuse matches full-parameter square models when scaling is correct; attention maps become more localized at error positions; and performance saturates near $d=2n$. These make the paper much more compelling than a pure benchmark paper.

---
## Weaknesses
My First concern is rigor. The main theoretical sections present “Conclusion 3.2” and “Conclusion 3.3,” but the paper gives more of an argument sketch and construction intuition than a formal theorem-proof treatment. I feel this somewhat heuristic, although the paper itself acknowledges that the embedding interpretation is heuristic, and the supporting evidence is indirect.

Second, the causal story is not fully disentangled. The gains seem to come from at least three changes at once: switching to a square circulant PCM, changing the effective input structure, and imposing cyclic parameter sharing. Table 3 helps by showing that sharing does not hurt relative to a full trainable square-circulant model, but it also suggests that a lot of the raw accuracy gain may already come from the circulant PCM design itself. I would have liked even cleaner ablations isolating each ingredient under matched compute and matched sequence length.

Third, the evaluation scope is somewhat narrow. The experiments are on BCH and PRM under AWGN, with all-zero-codeword training as in prior work. I would want more evidence on robustness: longer block lengths, other channel models, or at least a deeper equal-latency/equal-FLOP comparison.

Fourth, I would have liked a direct empirical comparison to the recent cyclic-code-specific Transformer work they cite, because the paper explicitly positions itself as more natural and theoretically grounded than that line. The current comparison set is strong, but that head-to-head would sharpen the novelty claim.

---

> ### Author Rebuttal · Authors · 2026-03-31
>
> Dear Reviewer,
>
> Thank you for your time and effort in reviewing our manuscript. After carefully considering the comments, we respond below.
>
> **Formal Structural Claims**
>
> Below, we present the proofs of Conclusions 3.2 and 3.3 in order to fill the potential theoretical gap.
>
> Proof conclusion 3.2.
>
> For any $1 \le j \le n$, by the definition of Error Correction Patterns,$c_i \in {\rm ECP}(v_j) \iff H_{i,j}=1$. Since the next row of $H$ is a right cyclic shift of the previous row, we have $H_{i,j}=1 \iff H_{i+1,j+1}=1,$ where the indices are taken modulo $n$. By the definition again, $c_i \in {\rm ECP}(v_j) \iff c_{i+1} \in {\rm ECP}(v_{j+1})$. Thus, $ {\rm ECP}(v_{j+1})=\{\,c_{i+1}: c_i\in {\rm ECP}(v_j)\,\}$, namely, ${\rm ECP}(v_{j+1})$ is the right cyclic shift of ${\rm ECP}(v_j)$.
>
> Proof conclusion 3.3.
>
> By the definition, $v_j \in $ $c_1$--VNs $\Longleftrightarrow$ $c_1 \in {\rm ECP}(v_j)$. By Conclusion 3.2, we have $c_1 \in {\rm ECP}(v_j) \Longleftrightarrow c_i\in {\rm ECP}(v_{j+i-1}) \Longleftrightarrow$ $v_{j+i-1} \in $ $c_i$--VNs, where the indices are taken modulo $n$. Thus $c_i$--VNs=$\{v_{j+i-1}: v_j \in c_1{\rm -VNs} \}$.
>
> Note that $v_i$--CNs=${\rm ECP}(v_i)$, the conclusion then follows from Conclusion 3.2.
>
> $c_j$--VNs $\in v_1$--VNs $\Longleftrightarrow$ $v_1 \in c_j$--VNs. By the above, $v_1 \in c_j$--VNs $\Longleftrightarrow v_i \in c_{j+i-1}$--VNs $\Longleftrightarrow c_{j+i-1}$--VNs $\in v_i$--VNs. Thus $v_i$--VNs=$\{c_{j+i-1}\text{--VNs}: c_{j}\text{--VNs} \in v_1{\rm -VNs}\}$.
>
>
> $v_j$--CNs $\in c_1$--CNs $\Longleftrightarrow c_1 \in v_j$--CNs. By the above, $c_1 \in v_j$--CNs $\Longleftrightarrow c_i \in v_{j+i-1}$--CNs $\Longleftrightarrow v_{j+i-1}$--CNs $\in c_i$--CNs. Thus $c_i$--CNs=$\{v_{j+i-1}\text{--CNs}: v_{j}\text{--CNs} \in c_1{\rm -CNs}\}$.
>
> We agree with the reviewer that the current exposition mixes a heuristic interpretation with a structural claim. Our intended logic is the following. Eq (2) shows that the $(i,j)$-th entry of $QK^\top$ is determined by $(\phi_i, \phi_j)$, which motivates viewing it as reflecting the relation between node $i$ and node $j$. Eq (4) further shows that each row/column of $W^Q$ linearly recombines these node-indexed quantities across all positions. Eq (2) and (4) motivate a node-aligned relational interpretation of the parameter matrix. Under this interpretation, we model the parameter matrix as being organized into four relation types ($V-V, V-C, C-V, C-C$). Conclusions 3.2 and 3.3 then rigorously show that the corresponding code-side relation families are closed under cyclic shifts induced by the circulant PCM. Therefore, the cyclic reconstruction rule used in Sec. 3.3 is an exact consequence of this structural prior.
>
> **Performance Gains**
>
> One point can be made clear: the performance improvement comes from the circulant PCM. Table 2 shows random square expansion helps only partially; Table 3 shows that under the same $n \times n$ circulant PCM, reuse matches the full-parameter model. Hence the main gain comes from the circulant prior, while reuse mainly preserves that gain with far fewer parameters. Finally, we have included additional and more detailed experiments. The relevant data and analysis can be found in our response to Reviewer ra2C, and we respectfully refer you to that response for further details.
>
> **Computational Complexity Concern and Longer Codes**
>
> You and several other reviewers have consistently raised related questions. Due to space limitations, the corresponding data and analysis have been included in our response to Reviewer hEPo/ra2C, and we respectfully refer you to that response for further details.
>
> **Comparison with Xiao et al. (2025)**
>
> Due to space limitations, for a theoretical comparison with Xiao et al. (2025), we kindly refer you to our final response to Reviewer ra2C. The table below presents a comparison in terms of performance and parameter count. Because the embedding dimension in Xiao et al. (2025) is subject to certain constraints, a direct comparison on the same scale is not feasible. Therefore, we use CrossMPT as the reference baseline.
>
> |Decoder||baseline1|||CrossMPT Xiao||||baseline2|||CrossMPT Ours|||
> |:-:|:-:|:-:|:-:|:-:|:-:|:-:|:-:|:-:|:-:|:-:|:-:|:-:|:-:|:-:|
> ||4|5|6|4|5|6|Parameter Reduction(%)|4|5|6|4|5|6|Parameter Reduction(%)|
> |BCH(63,36)|4.71|6.59|9.30|4.70|6.56|9.28|61.2|4.86|6.82|9.64|5.47|7.93|11.75|96.9|
> |BCH(63,45)|5.29|7.46|10.48|5.27|7.44|10.45|47.6|5.41|7.61|10.80|6.19|9.21|13.19|96.8|
> |BCH(63,51)|5.53|7.74|10.86|5.57|7.87|10.92|41|5.71|8.08|11.51|6.52|9.35|13.17|96.8|
>
> where baseline1 is Xiao's CrossMPT baseline, baseline2 is ours CrossMPT baseline. The results show that our method achieves clear advantages in both performance improvement and parameter reduction.
>
>
> Finally, we would like to sincerely thank you for your recognition of our work. We also greatly appreciate the time and effort you have devoted to reviewing our manuscript, and we look forward to your further feedback.

---

> > ### Author Rebuttal · Reviewer_yUf8 · 2026-04-04
> >
> > Thank the authors for the rebuttal with detailed explanations. The rebuttal solves my concerns and comments. The new results also clearly show their method's advantages. I will maintain my score.

---

### Official Review · Reviewer_hEPo · 2026-03-13

**Soundness:** 4
**Presentation:** 3
**Significance:** 3
**Originality:** 4
**Overall Recommendation:** 5
**Confidence:** 4

**Summary:**

This research addresses the tension between rigorous algebraic error correction and flexible neural transformer-based decoders. Traditional algorithms, such as the belief propagation or Berlekamp-Massey, rely on mathematical rigor but struggle with short block lengths or complex noise where maximum-likelihood decoding is NP-hard. Conversely, neural decoders like ECCT and CrossMPT achieve state-of-the-art performance by treating decoding as sequence-to-sequence problems. However, these models typically ignore inherent algebraic symmetries, resulting in high parameter counts and limited interpretability. Bridging this gap remains a key challenge for modern communication systems.

This work introduces drop-in circulant structural priors to integrate cyclic invariance into transformer architectures. By extending the parity-check matrix into an $n \times n$ circulant matrix, the model unifies error correction patterns across variable nodes, simplifying the learning task. Furthermore, a parameter reuse mechanism reconstructs weight matrices from minimal representative rows via cyclic shifts, achieving a 97% reduction in parameters while enhancing performance. Finally, the authors hypothesize that embedding dimensions serve as carriers for the Tanner graph’s topological adjacency, rather than abstract semantic features.

**Compliance With Llm Reviewing Policy:**

Affirmed.

**Final Justification:**

The authors' rebuttal convincingly addressed concerns regarding computational complexity and scalability, demonstrating that the circulant structural prior serves as a robust inductive bias rather than a simple architectural scaling trick. This research provides a compelling, interpretable framework for deploying high-performance decoders in resource-constrained environments. I maintain my score.

**Key Questions For Authors:**

- While parameters are reduced, the sequence length increases to $L=2n$. Given the $O(L^2)$ complexity of transformers, how does this affect FLOPs and energy consumption compared to standard methods?
- Can the representative row and block-wise shift mechanism be naturally extended to QC LDPC codes by applying shifts within sub-matrix blocks?
- Is the performance gain primarily driven by the increased redundancy or the unification of error correction patterns through symmetry?
- When $r > 1$, how are multiple representative rows initialized, and do they diverge into distinct features or remain highly correlated during training?
- Does the circulant prior influence error floors at high SNR? Furthermore, is standard cross-entropy sufficient, or is a specialized loss function required to achieve ML-level performance?

**Limitations:**

- The method is highly specialized for cyclic and PRM codes. Applying these circulant priors to non-cyclic codes, such as Polar or standard LDPC, can be detrimental to performance.
- While parameter memory is saved, the doubled sequence length $2n$ leads to quadratic complexity $O(L^2)$. This may result in higher latency and power consumption during real-time hardware inference.
- The requirement for square $n \times n$ circulant matrices limits flexibility in practical communication scenarios, such as puncturing or adaptive rate-matching.
- The current parameter reconstruction algorithm needs further optimization to ensure it can keep up with high-throughput hardware pipelines without becoming a bottleneck.

**Strengths And Weaknesses:**

* Soundness
- The paper is technically grounded in the explicit integration of cyclic invariance into the transformer architecture. By mathematically decomposing the attention mechanism ($QK^{\top}$), the authors provide a robust theoretical basis for their drop-in circulant priors.
- The soundness is reinforced by achieving a 97% parameter reduction while maintaining near-optimal performance. The R-select ablation study serves as a critical proof-of-concept, confirming that gains are derived from algebraic symmetry rather than simple architectural scaling.

* Presentation
- The manuscript follows an exemplary progression, identifying flaws in current models, proposing a mathematical solution, and validating it with exhaustive data.
- The use of attention heatmaps (Figures 5, 14, 15) effectively avoids the black-box trap by visually proving how the model localizes attention on error positions during the decoding process.

* Significance
- The drastic reduction in parameters enables high-performance neural decoders to run on resource-constrained FPGA/ASIC hardware, making them viable for 6G, IoT, and satellite communications.
- The research moves the field beyond brute-force learning toward topology-aware architectures, showing that neural networks can be designed to inherently understand the underlying algebraic structure of a code.
- As a modular set of structural priors, the method can be seamlessly integrated into existing models like ECCT or CrossMPT without requiring a total architectural redesign.

* Orignality
- Instead of merely tuning loss functions or masks, the work shifts the focus to the PCM itself. By designing the PCM as a circulant matrix, it transforms the code’s algebraic structure into a structural inductive bias, forcing the model to learn position-invariant strategies.
- The paper introduces a novel horizontal weight-sharing mechanism via block-wise cyclic shifts. This directly mirrors the cyclic properties of the code within the model's weights, representing a major conceptual leap from standard layer-wise sharing.
- The paper provides a highly original mechanistic explanation for the embedding space. It proves that embedding dimensions serve as carriers for Tanner graph topology, establishing a clear mathematical link between model capacity and the physical structure of the code.

* Weak Points
- The scaling strategy $d = 2rn$ lacks a rigorous first-principle derivation. The mathematical necessity of this specific ratio remains an empirical observation rather than a theoretically justified optimal choice.
- The validation is restricted to the idealized AWGN model. Further evidence is needed to prove that these algebraic biases remain robust in complex, non-Gaussian noise environments.

---

> ### Author Rebuttal · Authors · 2026-03-31
>
> Dear Reviewer,
>
> We would like to express our sincere gratitude for your time and effort in reviewing our manuscript. After carefully considering the comments, we provide our detailed responses to each point below.
>
> **Computational Complexity Concern**
>
> For this point, we provide the same response as that given to the other reviewers. Detailed results are reported in the table below. We agree that our method introduces a compute-performance trade-off. However, the absolute FLOPs increase only mildly, and considering the parallel computing capability of GPUs (see Table 5) as well as the substantial reduction in BER, we believe that this overhead is manageable. In addition, it can be seen that our method leads to a considerable reduction in memory usage; given its plug-and-play nature, this advantage is particularly meaningful for practical deployment. More importantly, as you have also noted, the deeper goal of this work is not merely to propose a new architecture, but to bring code-specific properties into Transformer decoding, thereby making the decoder more interpretable rather than purely black-box.
>
> ||CrossMPT||CrossMPT Ours||
> |-|-|-|-|-|
> |Codes|FLOPs(M)|Memory(G)|FLOPs(M)|Memory(G)|
> |BCH(63,36)|216.21|2.12|282.30|1.18|
> |BCH(63,45)|192.19|1.51|282.30|1.18|
> |PRM(127,64)|1863.42|4.67|2257.34|3.52|
> |PRM(127,99)|1683.94|3.95|2257.34|3.52|
>
> **Scalability**
>
> As the reviewer insightfully pointed out, the cyclic-equivalent reuse mechanism in our method can indeed be naturally transferred to the block-circulant structure of QC-LDPC codes. However, such a transfer mainly leads to a substantial reduction in the number of parameters, and does not by itself guarantee a performance improvement. How to extend the parity-check matrix of QC-LDPC codes in a principled and interpretable manner, so as to further translate this structural prior into performance gains, is an important direction for our future work.
>
> **Performance Gains**
>
> This concern was in fact already discussed in the original manuscript. Table 2 shows random square expansion helps only partially; Table 3 shows that under the same $n \times n$ circulant PCM, reuse matches the full-parameter model. Hence the main gain comes from the circulant prior, while reuse mainly preserves that gain with far fewer parameters. In addition, for POLAR(128,86), random expansion even results in a performance drop. Finally, we have included additional and more detailed experiments. The relevant data and analysis can be found in our third response to Reviewer ra2C, and we respectfully refer you to that response for further details.
>
> **The Case of $r>1$**
>
> For example, compared with the case of $r=1$, where only one representative row needs to be initialized, the case of $r=2$ requires initializing a pair of representative rows. This can be understood as a finer-grained characterization of features. However, these two representative rows do not have a shift relationship with each other. A more intuitive illustration can be found in Fig. 2 and Appendix A.
>
> **Error Floor and Loss Function**
> We did not observe an error floor within the $SNR=6$ range / within the lowest BER reached in our current experiments. We consider the channel condition at $SNR=6$ to be sufficiently good for this evaluation. Nevertheless, judging from the current trend, our method may lead to a lower error floor. In addition, we did not modify the loss function, and all other settings remain consistent with the original framework.
>
>
> Finally, we would like to sincerely thank you for your recognition of our work. We also greatly appreciate the time and effort you have devoted to reviewing our manuscript, and we look forward to your further feedback.

---

> > ### Author Rebuttal · Reviewer_hEPo · 2026-04-03
> >
> > Thank you for the detailed response and complexity analysis. The potential extension to QC-LDPC and the theoretical contribution of a 97% parameter reduction are impressive. These clarifications have fully addressed my concerns and I maintain my recommendation.

---

### Official Review · Reviewer_ra2C · 2026-03-13

**Soundness:** 2
**Presentation:** 3
**Significance:** 2
**Originality:** 3
**Overall Recommendation:** 2
**Confidence:** 4

**Summary:**

This paper proposes a method for incorporating cyclic code structure into Transformer-based decoders. The approach introduces a circulant parity-check matrix and cyclic parameter reuse to exploit cyclic symmetry, aiming to reduce model parameters and improve decoding performance.

**Compliance With Llm Reviewing Policy:**

Affirmed.

**Final Justification:**

The proposed method introduces an interesting approach to parameter sharing for cyclic codes by exploiting cyclic symmetry. However, this comes at the cost of increased computational complexity. Specifically, the attention map size grows as follows:

- ECCT: From $(2n-k)^2$ to $(2n)^2$, which is an increase by a factor of $\left(\frac{2n}{2n-k}\right)^2 = \left(\frac{2}{2-r}\right)^2$ (up to 4 times for high rate codes)
- CrossMPT: From $n(n-k)$ to  $n^2$, which is an increase by a factor of $\frac{1}{1-r}$ (10 times for $r = 0.9$)

Note that $r=k/n$ denotes the code rate. Furthermore, since the embedding dimension scales with $n$, the overall computational complexity increases from approximately $O(n^2)$ in ECCT/CrossMPT-style decoders to roughly $O(n^3)$ in the proposed method (I acknowledge the authors’ clarification in the rebuttal that the embedding dimension $d$ does not necessarily scale linearly with $n$ in practice). As also reflected in the rebuttal, this overhead becomes more pronounced for longer block lengths.

In the context of decoding, several factors are important, including decoding performance, computational complexity, memory usage, and parameter count. While the proposed method demonstrates clear advantages in parameter efficiency, it introduces a substantial increase in computational complexity. Although the rebuttal provides some evidence of reduced memory usage, the results are not comprehensive.

Moreover, improvements in decoding performance may also be achievable in existing methods by increasing model capacity (e.g., larger embedding dimensions or deeper architectures). Therefore, it remains unclear whether the proposed method provides a fundamentally stronger performance advantage beyond increased model expressiveness.

Another concern is that computational complexity and memory usage were not discussed in the original manuscript, despite being critical factors for practical decoding systems. While the rebuttal provides additional results, these appear partial and do not fully resolve the concerns regarding scalability and efficiency.

Finally, the applicability of the proposed method is limited to cyclic codes. In practice, widely used codes such as LDPC and Polar codes are generally not cyclic, which restricts the practical impact of the approach. As a result, the experimental evaluation is largely limited to BCH and PRM codes, which, while classical and well-studied, are less commonly used in modern applications.

For these reasons, while the paper has clear merits, I believe that the current version does not yet provide sufficient evidence to support its practical effectiveness.

**Key Questions For Authors:**

- Since the circulant PCM increases the syndrome length to $n$ and the input size to $2n$, could the authors report FLOPs, activation memory, and peak memory usage in addition to parameter counts? This would clarify the practical cost of the proposed design.
- The evaluation focuses primarily on relatively short codes. It would be helpful if the authors could report decoding performance, model size, and FLOPs for longer codes, in order to better understand the scalability of the proposed approach.
- The PCM used for the BCH visualization in Figure 5 appears to be non-systematic. Since prior work on ECCT and CrossMPT has reported performance sensitivity to PCM construction, could the authors clarify which PCM form was used for both the proposed method and the standard CrossMPT baseline in Figure 5?
- The authors state that *“our reasoning and method are intuitively more natural and theoretically more logical than prior work (Xiao et al., 2025).”* However, the precise technical distinction from Xiao et al. (2025) is not clearly explained. It would be helpful if the authors could explicitly clarify the methodological differences and how the proposed approach improves upon that work.

**Limitations:**

yes

**Strengths And Weaknesses:**

Strengths

- The paper reports significant large parameter savings, using as little as 1.91% and at most 4.16% of the original parameters, with an average of less than 3%, while still improving BER across several decoder families for cyclic codes.
- The use of an $n \times n$ circulant PCM to enforce full cyclic symmetry and unify error-correction patterns is an interesting architectural design choice in the context of Transformer-based ECC decoders.

Weakness

- A key design choice of the paper is to replace the original $(n-k)\times n$ PCM with an $n\times n$ circulant PCM, which increases the syndrome length to $n$ and the input length to $2n$. Consequently, the attention map size increases from $(2n - k)^2$ to $(2n)^2$.  Moreover, the proposed method sets the embedding dimension to to $d=2n$, whereas in ECCT-style decoders $d$ is typically fixed. Since Transformer attention complexity scales with both the sequence length and the embedding dimension, this design appears to increase the decoding complexity substantially. In particular, the complexity may scale roughly from $O(n^2)$ in the original ECCT-style design to approximately $O(n^3)$ in the proposed method. While the paper emphasizes parameter reduction, it does not fully quantify the resulting FLOPs, activation memory, or end-to-end computational cost.
- The proposed method introduces two substantial modifications: (i) replacing the PCM with a circulant $n\times n$ matrix, and (ii) applying cyclic parameter reuse across embeddings and network weights. It remains somewhat unclear whether the BER improvements primarily arise from the structural change in the proposed method or the increased decoding complexity.
- The proposed method is effective only for cyclic codes and not applicable to modern codes such as polar codes and LDPC codes.

---

> ### Author Rebuttal · Authors · 2026-03-31
>
> Dear Reviewer,
>
> Thank you for your insightful comments. We have carefully addressed your concerns below and would appreciate your reconsideration if you find them adequately addressed.
>
> **Computational Complexity Concern**
>
> We agree that our method introduces a compute-performance trade-off: the absolute FLOPs increase due to the longer sequence length, while memory usage decreases. Our additional comparisons also suggest that the performance gains cannot be explained solely by increased computation. Due to space limitations, the detailed FLOPs and memory are included in our first response to Reviewer hEPo, and we respectfully refer you to that response for further details.
>
> **Longer Codes**
>
> In the table below, we report longer-code results. Due to the inherent limitations of CrossMPT, the current code length is already close to its practical limit. The results show that our method still maintains a performance advantage at longer code lengths, while also involving certain trade-offs in computational complexity and memory usage, which have already been discussed in our previous response. Additional results are shown in Figs.4 and 7.
>
> |Decoder||CrossMPT|||||CrossMPT Ours||||
> |:-:|:-:|:-:|:-:|:-:|:-:|:-:|:-:|:-:|:-:|:-:|
> ||4|5|6|FLOPs(M)|Memory(G)|4|5|6|FLOPs(M)|Memory(G)|
> |PRM(511,256) d=73|2.88|3.29|4.04|1045.96|13.5|3.21|4.22|5.55|1251.31|6.72|
> |PRM(1023,638) d=31|3.26|3.75|4.36|784.39|25.76|3.55|4.67|5.86|1285.44|18.7|
>
> **Performance Gains (Key question 3 & Weakness 2)**
>
> ECCT and CrossMPT baselines use the original PCM from their original papers. Our method modifies the proposed model by replacing the original PCM with the $n\times n$ circulant square expansion. In addition, as reported in Table 2 of the manuscript, our method still achieves a clear performance lead when the computational cost is matched. Table 3 further shows that stable performance can be maintained with only a minimal number of parameters.
>
> In the table below, we provide comparative experiments against both a systematic PCM and a $(n+k)\times n$ PCM obtained by further randomly expanding the circulant PCM, while further cyclic expansion of the circulant PCM yields almost no additional performance gain. The results show that our method still achieves a substantial improvement over the systematic PCM. Moreover, compared with the $(n+k)\times n$ PCM, the additional performance gain is very limited, which may suggest that the cyclic equivalence exploited by our method already captures most of the useful check redundancy. Together with Table 2, Table 4, and Fig. 5 in the manuscript, these results consistently indicate that the performance gains of our method cannot be explained solely by increased compute, but from the structural changes it introduces into the decoder.
>
> |Decoder||CrossMPT systematic|||CrossMPT n+k|||CrossMPT Ours||
> |:-:|:-:|:-:|:-:|:-:|:-:|:-:|:-:|:-:|:-:|
> |Code/SNR|4|5|6|4|5|6|4|5|6|
> |BCH(63,36)|5.06|6.88|9.42|5.56|8.09|11.84|5.47|7.93|11.75|
> |BCH(63,45)|5.87|8.19|11.28|6.38|9.41|13.54|6.19|9.21|13.19|
> |BCH(63,51)|5.78|8.10|11.32|6.59|9.37|13.28|6.52|9.35|13.17|
> |PRM(127,99)|5.33|7.86|11.83|6.77|11.09|16.88|6.87|10.96|16.61|
>
> **Comparison with Xiao et al. (2025)**
>
> In Xiao et al. (2025), the authors observed a correspondence between the mask matrix and the cyclic structure in the Tanner graph. However, their formulation starts from the conventional $(n−k)\times n$ parity-check matrix, under which the cyclic equivalence of variable nodes still needs to be identified individually. For different codes, the corresponding cyclic equivalence patterns are entirely different, and there is no unified rule that can be directly applied. This makes the procedure not only cumbersome, but also ambiguous judgment, as reflected by their introduction of two separate criteria for identifying cyclic equivalence. In this sense, their sharing rule essentially amounts to recovering the residual equivalence that remains in the original representation.
>
> By contrast, our work starts from a different question that is, how to provide a better representation for cyclic codes. To this end, we first show that, under a circulant PCM, both the error-correction patterns of variable nodes and the four types of inter-node relationships can all be unified through cyclic-shift properties. As a result, the subsequent interpretation of embeddings, parameter reparameterization, and dimensional scaling all become direct consequences of this global symmetry, thereby forming a unified paradigm. This is precisely what we mean by saying that our approach is “more natural” and “theoretically more well-grounded”: we first restore the full symmetry and then impose parameter sharing accordingly.
>
> Moreover, for a more direct empirical comparison with Xiao et al. (2025), we refer you to our response to Reviewer yUf8, where we provide side-by-side quantitative results to make the differences between the two approaches more transparent.
>
> Thanks again for your feedback.

---

> > ### Author Rebuttal · Reviewer_ra2C · 2026-04-02
> >
> > While the parameter efficiency is impressive, the proposed design introduces a substantial increase in computational complexity that becomes more pronounced as the code length grows. This trend is also reflected in the PRM results: for PRM(511, 256), the computational cost increases by approximately 20%, whereas for PRM(1023, 638), the increase reaches about 64%, indicating that the overhead grows rapidly with the block length. Although parameter efficiency is an important advantage, the resulting increase in decoding complexity--from approximately $O(n^2)$ in the original ECCT-style design to $O(n^3)$ in the proposed method--represents a critical limitation, particularly for long-block-length regimes where computational efficiency is essential.

---

> > > ### Author Response · Authors · 2026-04-02
> > >
> > > We sincerely thank the reviewer for the timely feedback. While we acknowledge the increase in computational cost, we believe **this investment is highly worthwhile** for three key reasons: (i) it yields significant performance improvements alongside reduced memory usage; (ii) without our theory-guided model design, merely scaling computational complexity does not yield comparable performance; and (iii) the practical increase in computational overhead remains modest. Let us elaborate on these points below.
> > >
> > > First, within a plug-and-play Transformer-decoding framework, performance, computational complexity, and memory usage form a trade-off among three core metrics that are difficult to optimize simultaneously. While our method does incur some additional computational complexity, it delivers significant performance gains together with a substantial reduction in memory usage. For example, Table 1 and Fig. 5 in the original manuscript have already shown that our method brings substantial performance gains over the original model. In particular, in Fig. 7(o), for the longer $BCH(255,239)$ code, the gain reaches nearly three orders of magnitude. Moreover, as shown by the tabulated results in our previous response, our method can also significantly reduce memory usage. For PRM(1023,638), the baseline cannot even run on an RTX 4090 due to excessive memory consumption, in such cases, memory becomes a practical bottleneck. We therefore view the extra computation as a meaningful trade-off.
> > >
> > > Second, the gain is not simply purchased by more FLOPs. Through Table 2 in the original manuscript, we have shown that even under matched computational complexity, our method still maintains a substantial performance advantage. As discussed in our previous response under **Performance Gains**, the comparison with CrossMPT $n+k$ further shows that simply increasing computational complexity does not lead to additional performance improvement. This suggests that the observed improvement is mainly attributable to the structural change introduced by the cyclic prior, rather than to extra computation alone.
> > >
> > > Third, the practical overhead is milder than the worst-case asymptotic form may suggest. As shown in Table 5, when the parallel computation characteristics of GPUs are taken into account, the actual latency is not linearly related to the absolute number of floating-point operations. In addition, the $(O(n^3))$ statement corresponds to the setting where the embedding dimension scales linearly with the code length, namely $(d=2n)$. In practice, however, the embedding dimension in our method is scalable. For example, $PRM((511,256))$ uses $(d=73)$, and $PRM((1023,638))$ uses $(d=31)$. Therefore, although the overhead is certainly not negligible, the practical increase is much milder than the theoretical expression alone may suggest.
> > >
> > > We do not claim that the current method is optimal in all metrics. Rather, our claim is more specific: within a plug-and-play Transformer-decoding framework, the proposed cyclic prior yields a favorable BER/memory trade-off, and the observed gain is not explained solely by increased computation.
> > >
> > > Finally, beyond the empirical performance improvement, we believe that the more meaningful contribution of this work lies in introducing a certain degree of interpretability into the parameter matrix through the cyclic prior structure. In this way, the originally black-box decoding process is transformed into a relatively more observable and interpretable decoding logic, making the decoder no longer a complete black box.
> > >
> > > We sincerely appreciate your review and look forward to your further feedback.

---

### Decision · Program_Chairs · 2026-04-30

**Decision:**

Accept (regular)

**Comment:**

The paper introduces a novel way to incorporate the algebraic structure of cyclic codes into transformer-based decoders. As the reviewers note, the idea of using a circulant PCM and enforcing cyclic parameter reuse is orginal and well-motivated, and the reported parameter savings are impressive. However, as multiple reviewers have pointed out, this comes at the cost of increased computational complexity. The tradeoffs between parameter savings, decoding performance, computational complexity and memory usage need to be discussed in detail in the final version. One of the reviewers also commented that "the ablation experiments could be further strengthened to more definitively isolate the contribution of the structural prior from increased model capacity". Including the discussion and additional experimental results from the rebuttal phase will lead to a stronger and more complete final paper.